

# Functional classification of bioturbating macrofauna in marine sediments using time-resolved imaging of particle displacement and multivariate analysis

Stina Lindqvist[1], Johan Engelbrektsson[2], Susanne P. Eriksson[3] and Stefan Hulth[1]

[1]Department of Chemistry and Molecular Biology, University of Gothenburg, Göteborg, SE-412 96, Sweden
[2]SP Technical Research Institute of Sweden, Borås, SE-501 15, Sweden
[3]Department of Biological and Environmental Sciences – Kristineberg, University of Gothenburg, Fiskebäckskil, SE-451 78, Sweden

*Correspondence to*: Stefan Hulth (stefan.hulth@gu.se)

**Abstract.** Activities by macrofauna may drastically alter rates and pathways of reactions in surface sediments during early diagenesis and numerous experiments have been designed to quantify the importance of bioturbation by individual species and faunal communities for element cycling in these environments. It is increasingly recognized that functional traits of fauna are critical for element transformations. An example of such functional trait is the capacity for particle transport across horizontal and vertical gradients in environmental characteristics. The present contribution describes a general procedure for functional classification of fauna using multivariate analysis based on a suite of experimentally derived variables for particle reworking.

The relocation of fluorescently labeled particles (luminophores) added to surface sediments was quantified by side-view imaging during a two-week experiment incubating several common bioturbating species of marine benthic macrofauna: *Glycera alba*, *Nephtys incisa*, *Lipobranchius jeffreysii*, *Scalibregma inflatum* (Annelida), *Brissopsis lyrifera* (Echinodermata), *Abra nitida*, *Nuculana pernula*, and *Thyasira sarsii* (Mollusca) in thin glass aquaria.

Multivariate analysis revealed groups of species with similar mode of reworking based on reworking variables associated with quantity and time (bulk), as well as vertical distance (depth) of particle transport. Most pronounced effects on bulk transport were found in the *N. pernula*, *A. nitida* and *L. jeffreysii* treatments, while only a limited quantitative capacity to relocate particles was observed in the *T. sarsii* and *N. incisa* treatments. Although stochastic patterns were observed for some species, a prominent capacity for vertical transport of surface deposited particles was demonstrated for the annelids and *T. sarsii*.

From these results, three main groups of fauna with common reworking behavior were identified. *B. lyrifera*, *A. nitida* and *N. pernula* were species with only a limited effect on the vertical transport of particles. In contrast, while *N. incisa* and *T. sarsii* were able to relocate particles vertically, they shared a restricted capacity for bulk sediment transport. Despite high intraspecific variation, *G. alba*, *L. jeffreysii* and *S. inflatum* had the capacity for bulk and vertical transport of particles.

Despite the challenge to generalize species functionality and reworking capacity of benthic macrofauna, our results demonstrated that time-resolved high-resolution imaging of particle displacement, in combination with multivariate analysis, provides a general experimental tool for functional classification of benthic macrofauna.



## 1 Introduction

Bioturbated sediments are dynamic and heterogeneous environments with characteristics that vary over a wide range of temporal and spatial scales, in part related to functional traits of benthic fauna (Glud, 2008;Aller, 2014). While bioturbation constitutes a broad umbrella for displacements within sediments caused by activities of organisms (Berner, 1980), particle reworking can be defined as particle relocation within sediments induced by fauna (Kristensen et al., 2012). Particulate material may by e.g. feeding and defecation, burrowing and tube constructing activities be actively transported between reaction zones and redox boundaries. This transport, in combination with altered multi-dimensional transport geometries and patterns of diffusion within sediments and across the sediment-water interface has been observed to provide important controls during cycling of redox-sensitive elements crucial to the biosphere (e.g. C, N, P, Fe, Mn and S) (Aller, 1990;Canfield and Farquhar, 2009;Froelich et al., 1979). Consequences from particle reworking also include e.g. mixing of microorganisms and seed banks in the sediment (Piot et al., 2008), destruction of the stratigraphic record by homogenizing the sedimentary layer (Guinasso and Schink, 1975), as well as the burial and resurfacing of contaminants (Gilbert et al., 1994).

The approach to utilize concepts of species functionality and functional groups provides an intermediate level of complexity to study causal relationships between a taxonomical level of species and ecosystem function (Gerino et al., 2003). Species-specific reworking traits have been demonstrated to be important for ecosystem function *in situ*, in laboratory, and *in silico* experiments (Lohrer et al., 2004;Solan et al., 2004a;Norling et al., 2007) and the impact of rates and magnitude of particle displacements by functional traits of benthic macrofauna has been the focus of several studies (e.g. Gilbert et al., 2007;Braeckman et al., 2010;Hedman et al., 2011). Four major functional groups based on shared patterns of particle reworking have so far been defined; up- and downward conveyors, regenerators and biodiffusors (Francois et al., 1997). The upward conveyors are described as vertically oriented deposit feeders that ingest sediment at depth, and defecate at the sediment surface (Rhoads, 1974;Robbins et al., 1979). Such feeding mode causes an advective transport of sediment from depth to the sediment surface. Similarly, the downward conveyors are vertically oriented species that feed at the surface and defecate deep in the sediment (Smith et al., 1986). The regenerators excavate burrows by transferring sediment to the surface. The burrows are subsequently filled by either surface sediment infilling or collapsing of the burrow walls (Benninger et al., 1979;Gardner et al., 1987). Biodiffusors are organisms with activities that result in a constant diffusive transport of sediment, i.e. particles are transported in a random manner over short distances (Boudreau, 1986a;Robbins et al., 1979;Francois et al., 1997). In addition to these four main groups, the gallery diffusors are surface-active species that create galleries of burrows. Rapid, advective transport of particles occurs in the burrow system while the upper sediment layer is reworked in a diffusive manner. Initially suggested as a separate group (François et al., 2002), it has also been described as a subgroup of the biodiffusors, together with the epifaunal biodiffusors and surficial biodiffusors (Kristensen et al., 2012). In the classification scheme proposed by Solan et al. (2004a) epifauna form a separate group as do surficial modifiers, fauna with activities restricted to the sediment layer immediately below the sediment surface (<1-2 cm).

Classification of species from particle reworking is mainly based on laboratory experiments including one or a few species normalized to either constant biomass or abundance (e.g. Gilbert et al., 2007;François et al., 2002). Mode of



reworking is then generally determined by a few reworking variables modeled from the vertical redistribution of a particle tracer (Wheatcroft et al., 1990;Delmotte et al., 2008). Recently, functional classification was provided for more than 1000 benthic invertebrates to estimate the community bioturbation potential (BPc) (Queiros et al., 2013). Due to a paucity of data on the reworking behavior for many bioturbating species, classifications were not only based on published material but also expert knowledge and genetic similarity between species.

While it is important to determine functional groups of macrofauna related to complex ecosystem functions such as carbon and nitrogen cycling, a multitude of interactions between biotic and abiotic factors in benthic environments make this a challenge. To disentangle effects deriving from faunal activities, basic knowledge of processes that are directly and dominantly governed by bioturbating macrofauna, e.g. particle reworking, is essential (Meysman et al., 2006). Quantitative comparisons and classifications of particle reworking activities by single species of macrofauna provide a versatile tool to understand and predict feedbacks within and between macrofaunal communities, as well as between advective transport of particles by fauna and sediment biogeochemistry in natural environments.

Experimentally derived variables for sediment transport were in this study used to describe and determine functional modes of benthic fauna. Eight bioturbating species (*Glycera alba*, *Lipobranchius jeffreysii*, *Nephtys incisa*, *Scalibregma inflatum*, *Brissopsis lyrifera*, *Abra nitida*, *Nuculana pernula*, and *Thyasira sarsii*) were investigated by non-destructive time-laps imaging of luminophore displacement during two-week experiments. Patterns of particle relocation and functional modes of reworking for individual species were quantitatively and qualitatively evaluated by multivariate analysis based on experimentally derived variables for sediment transport. A suite of variables describing particle relocation was used to classify the species according to general modes of particle reworking behavior. While this contribution describes the classification of species based on particle reworking, the overriding principles are general and can be used for any aspect of bioturbation, e.g. irrigation and ventilation of pore water.

## 2 Materials and methods

### 2.1 Experimental overview

In total, particle reworking was studied in eight faunal and two control treatments during two experiments performed in 2008 and 2009 (n=4). During each experiment, four species were introduced to glass aquaria in mono-species additions. In 2008 one control aquarium was initially lost and therefore not further considered (i.e. $n_{tot}$=39).

Fluorescently labeled particles (luminophores) were added to the sediment surface and the walls of the aquaria were side-view imaged daily during two weeks. Particle reworking was quantified by variables extracted directly from the images or modeled from the vertical and horizontal redistribution of luminophores.

Reworking variables included, for example, the time-dependent 2D distribution of luminophores (Lindqvist et al., 2013) and the maximum penetration depth of luminophores (MPD) (Maire et al., 2006). The 2D images were also collapsed into 1D vertical tracer distributions from which the gallery-diffusor model could be evaluated (François et al., 2002).

The species investigated are common in surface sediments from the Skagerrak and have from previous investigations been assigned to different functional groups of benthic macrofauna in terms of feeding behavior, mobility, and





particle reworking modes (Table 1). They were selected as model species in this experiment due to their wide distribution, common occurrence and expectance of having different functions with respect to modes of particle transport.

**2.2 Sampling and experimental set-up**

In 2008 and 2009, surface sediment from the Gullmarsfjord (north-eastern North Sea; 58° 17.35 11° 30.91 and 58°17.37 11° 30.72, water depth 78-81 m) was sampled by an Olausson box corer (Table 2) and used in two separate experiments. The sediment was vertically sectioned in ~3 cm depth intervals from the sediment surface to 15 cm depth. The sediment from each layer was sieved separately (1 mm mesh size) to remove macrofauna and larger debris. Thin glass aquaria ($16 \times 33 \times 1.4$ cm$^3$) were consecutively filled with the sieved sediment in corresponding

order to reconstitute vertical gradients of sediment properties (Michaud et al., 2010). The aquaria were placed in a water-filled plastic container and the overlying water was treated with $N_2$ during 7 days (2008) or 4 days (2009), as an additional precaution to remove any fauna remaining in the sediment through asphyxiation. Eventual fauna and tube constructions visible on the sediment surface were manually removed. The aquaria were after this period of anoxia continuously aerated by aquaria pumps and incubated in a temperature-controlled room at 10 °C. Water in the

aquaria was renewed every second day with unfiltered water from the Gullmarsfjord (depth = 32 m, S ~ 33).
Fauna was sampled in sediments from the Gullmarsfjord at water depths of 33-97 m using a benthic sledge (Warén). Animals were placed in containers with unsieved sediment from the sampling locations under continuous water flow. The fauna was added to the experimental aquaria a week later, i.e. about four weeks after the aquaria were initially filled with sediment. The organisms were wet weighted (ww) and their individual volume determined before the

addition of one individual to each aquarium, corresponding to an abundance of 460 ind. m$^{-2}$. Most animals buried within 5 minutes (60%), 75% were buried within 30 minutes and the final animal disappeared from the sediment surface after 6 hours (*Thyasira sarsii*). *Brissopsis lyrifera* and *Thyasira sarsii* were the last species to bury into the sediment. The biomass varied between 26-290 g m$^{-2}$ and the biovolume between 23-230 mL m$^{-2}$ in the aquaria (Table 3).

**2.3 Optical instrumentation and image analysis**

Luminophore distributions were quantified by consecutive side-view imaging of fluorescence during the experimental period (Fig. 1). The fluorescently labeled particles (63-125 μm, $\lambda_{ex}$ = 500 nm, $\lambda_{em}$ 602 nm) were added 1-2 weeks after fauna was introduced to the aquaria (Table 2). Luminophores were mixed with ≈ 25 mL of overlying water into a turbid suspension that was carefully dispensed into the overlying water of the aquaria with a

polypropylene syringe to a final density of 265 g m$^{-2}$. The excitation light source was a LED (Luxeon 570 nm) and fluorescence emission was captured by a Canon EOS 350D digital camera with a bandpass filter (Thorlabs; 610 nm ± 10nm) mounted on-line with the CMOS chip. To minimize interfering light and reduce effects from a corrosive atmosphere, the camera and light source were placed inside a black box of PVC made in-house. High throughput sampling was accomplished by custom-built high precision rotary sample turrets (Fig. 1). Exchangeable base plates

and detachable sample cells provide flexibility for a large variety of applications (Strömberg et al., 2009). Aquaria



were positioned in the 24 sample cell turret, powered by a servomotor rotation stage (Thorlabs CR1-Z7E) controlled through a stand-alone program supplied with the device. Image acquisition was controlled from an adjacent room by computer. In total, 21 images were taken of each aquarium, of which 10 images were captured during the first 3 days. After 3 days, images were acquired once a day. The total tracer incubation period was 14 days.

All images were captured in the RAW (.CR2) format with a 3456×2304 pixel resolution (corresponds to ~8 M pixels). Images were saved in Exef-TIFF (8 bit, Digital Photo Professional 2.2.0.1) and processed in Matlab7.5 (R2010B). A 2100×2100 image was used for image processing and the effective pixel size was ~75 μm pixel$^{-1}$. The red layer of the CMOS chip was used for quantification and images were thresholded. The threshold was set to maximize visualization of luminophores while minimizing e.g. reflection and scattering of the incident light. In the

resulting binary matrix, the sediment surface was determined. Prior to processing and quantification of the 2D reworking variables, images were re-aligned (pixel by pixel) by using the plugin TurboReg (Thevenaz et al., 1998) in the software ImageJ (Version 1.45). Because some luminophores were attached to the glass wall in the water phase, pixels above the sediment-water interface were set to 0. In addition, the time-dependent compaction of the sediment was compensated for by adjusting the image matrix upwards at the same rate as compaction affected the sediment.

For the 1D variables, images were pre-treated by flattening the sediment surface. Each column of pixels was individually adjusted so that pixels corresponding to the sediment surface were in the first row (Solan et al., 2004b).

## 2.4 Quantification of reworking variables

The 2D reworking variables 2D redistribution (2Dredist) and transport rate (Rate) were calculated according to Lindqvist et al. (2013) and originate from the basic principles of the optical reworking coefficient (ORC) (Gilbert et

al., 2003). In short, the binary matrix from one image was subtracted from that of the previous image. The resulting matrix consisted of zeros where no transport had occurred, +1 for appearance and -1 for disappearance of luminophores. The integrated sum of the matrix, using absolute values, was used as a quantitative measure of the total relocation of luminophores. The 2D redistribution of particles was calculated by the binary image obtained at time t ($\mathbf{M}_t$), subtracted by the binary image from the start of experiments ($\mathbf{M}_0$). The daily transport was calculated by

the binary image obtained at time t ($\mathbf{M}_t$), subtracted by the binary image obtained the previous day ($\mathbf{M}_{t\text{-}24\text{ h}}$). The transport rate of particles, slightly modified from that described in Lindqvist et al. (2013), was calculated from the sum of the daily transport ($\Sigma T_{daily}$) divided by the number of experimental days ($\Sigma T_{daily}/14$).

Another 2D variable was extracted from the last image in the time series (Rugosity; Rug). Rugosity was calculated as the sum of distances between the shallowest and the deepest luminophore in the aquaria over the image (Murray et

al., 2014).

The pixels of vertical layers including luminophores were summarized and assigned to the midpoint of each layer to reduce the 2D image to a 1D vertical profile. From these profiles, variables describing the relative fraction (%) of luminophores transported beneath 0.5 (0.5cm), 2 (2cm) or 4 cm (4cm) were extracted. 1D tracer distributions were also used to quantitatively evaluate the particle transport by the gallery-diffusor model, which approximates the

diffusive mixing by the biodiffusion coefficient (Db) and the rapid transport over long distances by the non-local transport coefficient, r (François et al., 2002). The model is time-and space dependent and employs ordinary



differential equations to minimize a weighted sum of squared difference between the observed and modeled tracer concentrations with sediment depth.

The maximum penetration depth of luminophores (MPD) was calculated from the deepest pixel row containing luminophores (Maire et al., 2006). To exclude noise, the limit of quantification was set to 3 pixels in 2008 and 8

pixels in 2009. The limit was determined from the camera response in deep sediment layers of the control cores with no luminophores (i.e. background). The sediment depth at which animal activity was observed from photos of the aquaria walls taken under normal light conditions was also determined (Burrow).

### 2.5 Statistical analysis

A principal component analysis (PCA) was performed in Simca (v. 14.0, Umetrics AB, Sweden) to identify groups

with similar particle reworking behavior and to evaluate the most relevant reworking variables to be included in models of functional classification. Seven variables for particle reworking were included in the final model. Prior to analysis, variables were assessed by graphical exploration of box plots (Quinn and Keough, 2002). Where appropriate, variables were log(x) or log(x+1) transformed in order to fit the requirements for PCA The variables were mean centered and scaled to unit-variance. An initial evaluation showed different behavior for the 2009

experiment compared to 2008, and the two sets were therefore modeled separately. The models were diagnosed by $R^2$ and $Q^2$, representing their ability to describe ($R^2$) and predict ($Q^2$) the variation in the data (Eriksson et al., 2006). A cluster analysis ($n_{groups}$=3) was performed on the results from the PCA in Matlab. Centroids were determined by a two-phase iterative algorithm that minimized the sum of point-to-centroid distances, summed over all the clusters by squared Euclidean distances. Where appropriate, general results were presented as mean ± standard deviation.

## 3 Results

### 3.1 Visual observations

The bivalves *Abra nitida* and *Nuculana pernula* reworked a large portion of the sediment surface. Mounds (larger in the *N. pernula* treatments) were created across the luminophore layer initially deposited on the sediment surface. According to visual observations of the aquaria *A. nitida* reworked the sediment to 1.5-2.5 cm and *N. pernula* down

to 1-2 cm depth. At the termination of the incubation *A. nitida* was recovered at 1-2.5 cm depth and *N. pernula* at 1.5-3.5 cm depth. Traces from activities by the echinoderm *Brissopsis lyrifera* were visible down to 1 cm depth and organisms were recovered at 2.5-3 cm depth. While the roughness of the sediment surface increased for this species, no mounds were evident. *Thyasira sarsii* created its characteristic sulfide-mining burrows shaped like tree roots between the depths 3-5 to 9-10 cm. Specimens were recovered at 3-6 cm. Orange-colored flocculent organic material

developed on the sediment surface immediately adjacent to the burrow opening, possibly due to formation of hydrated iron (Dando et al., 2004). Except from the formation of this flocculent organic material the sediment surface was visually undisturbed. The annelid *Nephtys incisa* created burrows visible 5-10 cm deep on the aquaria walls and depth of recovery was 6-15 cm. The sediment surface was only disturbed where the burrows reached the surface. *Glycera alba* created the most extensive gallery of burrows visible on the aquaria wall down to 12-15 cm





and animals were recovered between 7 and 15 cm. Evidence of disturbance of the sediment surface was observed in some of the replicates. *Lipobranchius jeffreysii* created burrows down to 8-15 cm and were recovered between 5 and 11 cm depth. For some replicates, mounds were created across the luminophore layer. *Scalibregma inflatum* burrowed to the bottom of the aquaria (15 cm) and the sediment surface was disturbed in some replicates. Depth of

recovery was 13-15 cm.

### 3.2 Distribution and maximum depth of luminophores

Relocation of luminophores from the sediment surface deeper into the sediment was observed in all faunal treatments, although there was a wide spectrum in qualitative and quantitative patterns of sediment transport (Fig. 2, Table 4). For the 2008 controls, 100 % of the luminophores were observed in the 0–0.5 cm sediment layer. In 2009,

99.7-100 % of recovered particles were found in the surface layer in three of the controls. In one control aquarium only 95.0 % was found in the surface layer, suggesting a transport of luminophores from the sediment surface. As visual inspection of individual aquaria confirmed traces of fauna, this aquarium was excluded from all further analyses.

Downward transport of the tracer over time was quantified by the maximum penetration depth of luminophores

(MPD) (Fig. 2). For the *N. incisa* treatment, luminophores were frequently observed below 10 cm sediment depth. Occasionally, particles were observed at ~15 cm depth in the sediment. A significant downward transport of surface particles was also sporadically observed for individual treatments of *T. sarsii* (~ 9 cm), *L. jeffreysii* (~ 10 cm) and *S. inflatum* (~ 10 cm). Patterns of particle redistribution suggested that *G. alba* transferred particles from the sediment surface to ~ 5 cm depth. At specific sampling occasions and in individual aquaria, luminophores were observed at ~

9 cm sediment depth at the end of the incubation period. Luminophores in the *A. nitida* treatment were mainly observed within the first cm of the sediment. In one aquarium, however, particles were observed at ~ 5 cm depth. Luminophore distributions further indicated a limited capacity for *N. pernula* ($MPD_{max}$=2.2 cm) and *B. lyrifera* ($MPD_{max}$=2.3 cm) to vertically relocate particles from the sediment surface to deeper sediment layers. In these treatments, there was also a clear tendency of similar patterns of particle reworking between individuals within the

species.

*N. pernula*, *L. jeffreysii* and *A. nitida* transported the largest fraction of particles beneath the sediment surface (0.5cm, Table 4). The bivalves were not able to relocate larger quantities of particles to deeper sediment layers. *L. jeffreysii*, *G. alba* and *N. incisa* contributed most to particle relocation beneath 2 and 4 cm (2cm and 4cm, Table 4).

### 3.3 Db and r from imaging

The highest values of Db were found in treatments with *N. pernula* (1.3 ± 0.82 cm$^2$ yr$^{-1}$), *A. nitida* (0.68 ± 0.79 cm$^2$ yr$^{-1}$) and *L. jeffreysii* (0.52 ± 0.48 cm$^2$ yr$^{-1}$) (Fig. 3). Excluding the controls, lowest values of Db were found in treatments with *S. inflatum* (0.22 ± 0.24 cm$^2$ yr$^{-1}$), *N. incisa* (0.24 ± 0.28 cm$^2$ yr$^{-1}$) and *T. sarsii* (0.26 ± 0.32 cm$^2$ yr$^{-1}$). Although there was a large variability between replicates and there were only few aquaria where r could be quantified, indications of deep particle injection and non-local transport (defined by r) were observed for *N. pernula*



(3.5 ± 4.3 yr$^{-1}$) and for *L. jeffreysii* (1.1 ± 1.3 yr$^{-1}$). The biodiffusion coefficient and the non-local transport coefficient were below detection in the control aquaria.

### 3.4 2D reworking variables

The temporal evolution in distribution patterns of luminophores following bioturbation activities was quantified by

2D particle redistribution (Fig. 4). The 2D redistribution describes the change in the particle distribution over time relative to the initially deposited layer of luminophores. Over the two-week experimental period there was a linear to logarithmic increase in particle redistribution with time of incubations (Fig. 4). Although patterns were different between treatments, the largest increase with time was in general observed during the first ~two days. The largest quantitative capacity for bulk particle transport was observed in the *N. pernula* treatment (Fig. 4). While the 2D

redistribution reached ~ 2 cm$^2$ in three replicates, this variable peaked at ~ 7 cm$^2$ in one aquarium, i.e. 2-20 times the maximum quantity of luminophores relocated in any other aquaria. Comparing the mean value between species, largest quantities of luminophores were relocated by *N. pernula* (3.2 ± 2.3 cm$^2$) and *A. nitida* (2.2 ± 1.6 cm$^2$). Although there were individual traits within species, the lowest particle redistribution in faunal treatments was observed in the *N. incisa* (0.5 ± 0.2 cm$^2$) and the *G. alba* (1.1 ± 0.5 cm$^2$) treatments. In the controls, an initial period

of particle redistribution occurred during the first two days, after which the distribution remained relatively constant for the rest of the experiment with final values reaching 0.4 ± 0.1 cm$^2$ (2008) and 0.7 ± 0.2 cm$^2$ (2009).

The daily particle transport quantifies the number of particles that are relocated each day (Fig. 5). There was a general trend of similar activity over time for several species. However, sporadically there were intense reworking activities in specific aquaria, e.g. in the *N. pernula B. lyrifera* and *S. inflatum* treatments. The rate of particle

transport describes the mean intensity of reworking during the experiment (Fig. 3). Similar to observations made for the 2D redistributions, rate of particle transport suggested that the most pronounced transport of bulk sediment was observed in treatments with *N. pernula* (0.98 ± 0.91 cm$^2$ d$^{-1}$), *L. jeffreysii* (0.56 ± 0.31 cm$^2$ d$^{-1}$), and *A. nitida* (0.50 ± 0.36 cm$^2$ d$^{-1}$). The least capacity for bulk transport was observed for the *N. incisa* (0.17 ± 0.03 cm$^2$ d$^{-1}$) and *T. sarsii* (0.23 ± 0.09 cm$^2$ d$^{-1}$) treatments.

Highest values of rugosity were found in treatments with *L. jeffreysii*, *G. alba* and *N. pernula* while *T. sarsii*, *A. nitida*, while *N. incisa* had the least influence on rugosity (Table 4).

### 3.5 Multivariate analysis of bioturbation variables

The multi-variable approach using a suite of bioturbation variables well described the data set from 2008 (R$^2$ = 0.91 and Q$^2$ = 0.75). Multivariate modeling also described data well from the experiment in 2009, but the predictive

capacity was lower (R$^2$ = 0.86 and Q$^2$ = 0.60).

In the multivariate models, the applied particle reworking variables generally separated in two groups, one for each principal component (Fig. 6). The first principal component (PC1) was dominated by variables describing bulk sediment transport (2Dredist, Rate and 0.5cm). The second principal component (PC2) was dominated by variables describing depth of particle relocation (Burrow and MPD). Rugosity was positioned intermediately between the two





principal components. This indicated that rugosity explained variation in both bulk (PC1) and vertical transport (PC2) of particles.

Both the biodiffusion coefficient (Db) and the non-local transport coefficient (r) were removed from the final multivariate models due to generally low information content in r and low contribution to the models for Db. The variables that described transport to deeper sediment layers derived from the 1D distribution of luminophores (2cm and 4cm), clustered with the Burrow and MPD. They were, however, removed due to a generally low contribution to the multivariate models.

Overall, the control aquaria were tightly grouped and separated from the faunal treatments (Fig. 6). Treatments of macrofauna were also separated in two major groups. Effects from bioturbation by *N. incisa* and *T. sarsii* appeared more tightly coupled to a relocation of particles to deeper sediment layers. In contrast, consequences from particle reworking by *N. pernula* and *A. nitida* seemed more associated with bulk sediment transport.

In the 2009 experiments, species of macrofauna were not grouped and separated as clearly as during 2008. However, the *B. lyrifera* treatments appeared associated with bulk sediment transport, with only minor effects on the vertical sediment transport.

The cluster analysis separated functional groups from controls (Fig 7). Aquaria with low reworking activities by fauna were included in the control groups. *N. incisa* and *T. sarsii* treatments grouped together, as did *A. nitida* and *N. pernula* treatments. In the 2009 experiment, one of the clusters was made from treatments where transport of particles to deep sediment layers was the dominant transport mechanism. This cluster included a majority of the *S. inflatum* aquaria as well as two *L. jeffreysii* aquaria. The cluster analysis groups the treatments according to Euclidian distance and the analysis is insensitive to the intensity trajectory created by the reworking variables. As a consequence, the other cluster included treatments of fauna that affected bulk sediment transport, as well as a combination of bulk transport and depth of particle relocation.

## 4 Discussion

### 4.1 Experimental variables for mechanisms of particle transport

This study presents a classification scheme based on comparing treatments by a principal component analysis performed on a wide array of measured variables for particle reworking (Table 4, Figs. 2, 3, 4 and 6). Such analysis reveals the integrated behavior of the sediment system by sorting variables that measure the same fundamental transport mechanism in clusters and replacing them with a principal component. Based on obtained results there appears to be two main principal components that separate the fauna into different groups; the bulk quantity of particles that are relocated over time (PC1) and the vertical distance particles are displaced (PC2) (Fig. 6).

An important concept in the general classification of fauna according to particle reworking mode has been the distinction between local and non-local mixing (Meysman et al., 2010). The basic principles derive from the biodiffusion model that is underpinned by assumptions of local (diffusive-like) particle displacements (directionally random, very frequent and small particle displacements) (Boudreau, 1986a). Non-local (advective) mixing, on the other hand, allows for long distance transport over short time scales (Boudreau, 1986b;Boudreau and Imboden,



1987;Francois et al., 1997). The biodiffusion concept has been further refined with the development of random walk models for particle reworking (e.g. Boudreau, 1989;Wheatcroft et al., 1990;Meysman et al., 2010). Random walk models track individual particles and quantify Db with waiting time, jump length, and direction of particle movement, as either known and fixed variables, or known frequency distributions (Meysman et al., 2010;Wheatcroft et al., 1990). These frequency distributions provide an individual "fingerprint" for communities or species and can theoretically be used to group bioturbators into functional groups (Meysman et al., 2008;Bernard et al., 2012). Random walk models are more general than the classic biodiffusion model in that they are also able to describe non-local particle transport over short time scales, while they converge to diffusive models over long time scales. However, they are still restricted to modes of reworking that generate random mixing of particles (Meysman et al., 2010).

In general, results from the multivariate analysis in the present study support a divergence between frequent/infrequent relocation as well as between short/long transport distances (Fig. 6). Without being restricted to specific assumptions of transport (e.g. infinitely small and frequent step length), variables associated with PC1 described bulk relocation of particles over a fixed time – that is intensity, or frequency of mixing, while variables associated with PC2 described depth of particle relocation.

By the general convention of diffusive mixing, species associated with high bulk transport and shallow depth of particle relocation fit the criteria of a diffusive-like mixing mode (e.g. *N. pernula*; e.g. Fig. 2 and 4), while species associated with low bulk transport and shallow depth of particle relocation induce a diffusive-like transport over long time scales (e.g. *B. lyrifera*; e.g. Fig. 2 and 4). Similarly, species that generate a distant vertical relocation of particles over short time scales induce a non-local transport (e.g. *N. incisa;* e.g. Fig. 2 and 4).

The gallery-diffusor model describes particle reworking by species of macrofauna that generate both a surficial diffusive-like transport as well as an advective transport of luminophores to deeper sediment layers (François et al., 2002). By applying the gallery-diffusor model to a range of species with different modes of particle transport, species with only a biodiffusive-like mode were expected to be discriminated from species with a non-local mode of particle transport. However, such general classification was not obvious from the present data set (Fig. 3). For example, quite in contrast to other reworking variables associated with depth of particle relocation, it was possible to quantify non-local transport in the *N. pernula* treatment. In addition, r was close to or below detection in treatments including species to which a non-local mode of particle transport previously has been ascribed (e.g. *S. inflatum*; Gilbert et al., 2007). Despite the ability to rapidly relocate particles to deeper (MPD>5 cm) sediment layers , r was close to or below detection for *N. incisa*, T. *sarsii*, *G. alba* and *S. inflatum*. Thus, it appeared challenging to model gallery-diffusive mixing based on 1D particle distributions obtained from side-view imaging (Lindqvist et al., 2013;Maire et al., 2006;Dorgan et al., 2006).

### 4.2 Classification of macrofauna

Three of the investigated species (*Brissopsis lyrifera*, *Abra nitida* and *Nuculana pernula*) shared the common property of mainly affecting variables for bulk sediment transport, with only minor impact on variables that describe depth of particle relocation. In previous investigations they have been ascribed either to the biodiffusor group (*B.*



*lyrifera, A. nitida*) or to the surficial modifier group (*A. nitida, Nuculana minuta*), observations that were in agreement with this study (Table 1, Fig. 6). Sea urchins move through the sediment in a bulldozing mode and the subsurface deposit-feeder *B. lyrifera* move through the sediment with a slow rocking motion, displacing large volumes of sediment (Hollertz and Duchene, 2001;Duchene and Rosenberg, 2001). *N. pernula* is a burrowing, semi-

mobile and subsurface feeding bivalve. While its reworking activities are not well documented, *N. minuta* (of the same genus) has been assigned as a surficial modifier with slow, free movement through the sediment matrix (Queiros et al., 2013). Reworking by the genus *Abra*, on the other hand, has been studied in great detail, (e.g. Grémare et al., 2004;Bernard et al., 2012). The surface deposit feeder *A. nitida* has been categorized as a biodiffusor in studies using luminophores as tracer for particle transport during single-species sediment-water incubations (Maire

et al., 2006;Gilbert et al., 2007). Arguments for classifying *A. nitida* as a biodiffusor include observations of the animal reworking the upper few centimeters of the sediment without creating any particular structures due to randomly prospecting the sediment surface by the siphons in combination with lateral movements of the shell (Bellas et al., 2006;Maire et al., 2006). The reworking activity varies depending of food availability, from only small lateral oscillations of the siphon, to periodically intense events corresponding to shell displacement (Grémare et al., 2004).

*Abra* are able to transport luminophores both up- and downwards in the sediment (Gilbert et al., 2007;Bernard et al., 2012), observations which would further support the classification of *A. nitida* as a biodiffusor. However, other results from these studies indicated that such classification may not be appropriate for all situations. Bernard et al (2012) also found lateral and vertical heterogeneity in particle displacements by *A. alba* using the random-walk model approach (i.e. non-random mixing). Also, in Gilbert et al. (2007), a non-local transport was modelled

following *A. nitida* reworking activities and observations of subsurface peaks in tracer distributions were found in Maire et al (2006). *A. nitida* has been recovered relatively deep in the sediment (~10 cm) under experimental conditions (Norling et al., 2007;Gilbert et al., 2007).

Despite these indications of non-diffusive transport and potentially deep burial of tracers by *A. nitida*, observations from this study support the classification of these three species as either biodiffusors or surficial modifiers.

Based on results from the cluster- and multivariate analysis *Nephtys incisa* and *Thyasira sarsii* were grouped together (Fig. 7). These species shared the common trait of mainly affecting variables related to depth of particle relocation while having a limited capacity for bulk sediment transport (Fig. 6). The predatory annelids *Nephtys* create galleries of burrows, generally within the top 10 cm of the sediment (Diaz and Schaffner, 1990;Fauchald and Jumars,

1979;Michaud et al., 2010). While *Nephtys* are usually considered carnivores feeding on small invertebrates (Fauchald and Jumars, 1979), *N. incisa* has also been described as a mobile subsurface deposit-feeder (Sanders, 1960). *T. sarsii* is a semi-mobile, subsurface feeding bivalve. The chemosymbiotic *T. sarsii* requires reduced sulfur and energetically favorable oxidants for maintenance of the symbionts on which they feed (Dufour and Felbeck, 2003). *N. incisa* has previously been described as a biodiffusor creating transient burrows (Michaud et al.,

2010;Queiros et al., 2013), based on observations of *N. caeca* rearranging particles in a homogenous fashion in the upper first centimeters of the sediment as a result of a free movement (Piot et al., 2008). *T. sarsii* has been described as a downward conveyor (Queiros et al., 2013). Observations in the present study are somewhat in contrast to these classifications as neither of the species displayed the behavior associated with biodiffusors (random particle transport




over short distances) or downward conveyors (vertically oriented species that feed at the surface and defecate deep in the sediment). Both *N. incisa* and *T. sarsii* created a burrow system during the course of the experiment. Seemingly they remained in the same part of the aquarium instead of prospecting the whole area available. Burrows were prolonged and a few collapsed with time, an observation in agreement with previous studies of *T. sarsii* (Dufour and

5 Felbeck, 2003). This reworking behavior has also been described previously for *Nephtys* whose burrows are not permanent. Abandoned burrows can be filled by sediment or organic matter accumulating, or reoccupied by fauna (Michaud et al., 2010). Individuals of *Nephtys caeca*, have been observed to move freely through the sediment matrix, but preferred to return to abandoned burrows (Piot et al., 2008). While the genus *Nephtys spp.* has been described as active with a capacity to burrow rapidly through crack propagation (Clark, 1962;Fauchald and Jumars,

1979;Dorgan et al., 2006), movement patterns are also characterized by long phases of inactivity (Piot et al., 2008). Other studies of *Nephtys* have also concluded a comparatively low particle reworking activity (O'Reilly et al., 2006;Braeckman et al., 2010). The reworking activities by *Nephtys hombergii* was masked by activities by both *Amphiura filiformis* and *Leptopentacta elongata* in the study by O'Reilly et al (2006). The low activity of *Nephtys sp.* in Braeckman et al. (2010) was suggested to be attributed to sediment devoid of prey due to freezing the sediment

before animal introduction. In the study by Michaud et al (2010), burrow depth and burrow density was found to be correlated to the distribution and abundance of fresh, reactive organic material in the sediment. In parallel, the length and number of burrows created by thyasirids have been correlated with the concentration of hydrogen sulfide in the sediment (Dufour and Felbeck, 2003).

Species that relocate particles fast to deep sediment layers on a short time-scale (i.e. non-local transport) are

20 sometimes considered as biodiffusors as this behavior is expected to produce diffusive-like mixing after a sufficient number of bioturbation events (Meysman et al., 2010). For example, *Nephtys hombergii*, *Alitta virens*, *Hediste diversicolor*, and *Amphiura filiformis* have all been described as biodiffusors, while they are also known to induce a non-local particle transport (Murray et al., 2014;Queiros et al., 2015;Gilbert et al., 2007). An alternative classification for this type of reworking behavior would be gallery-diffusion. However, the species *N. incisa* or *T.*

*sarsii* did not demonstrate a surface feeding behavior associated with gallery-diffusors. Therefore, these two species were not expected to produce diffusive-like mixing in the surficial layers. The present study thus suggested that *N. incisa* and *T. sarsii* were better described as gallery-builders with a limited surface activity and with minor influence on bulk sediment transport.

It was a challenge to describe and classify the reworking behavior by the three annelids *Glycera alba*, *Lipobranchius jeffreysii* and *Scalibregma inflatum*. Several reworking variables from both main principal components were expressed in the PCA with high intraspecies variation (Fig. 6). In accordance, the cluster analysis placed treatments of these species in two groups (Fig. 7). In one group, variables describing bulk transport were the most influential, and in the other, variables describing depth of particle relocation were the most important. *G. alba* are carnivores that

form semi-permanent burrow systems in soft substratum (Ockelman and Vahl, 1970;Fauchald and Jumars, 1979). They create a complex burrow system with several openings, through which they track moving prey (Fauchald and Jumars, 1979). The annelids *L. jeffreysii* and *S. inflatum* are subsurface deposit-feeders considered as active burrowing species of macrofauna. *S. inflatum* is normally observed in muddy or mixed sediment strata where they



construct galleries down to 4–5 cm supported by mucus. Occasionally they have been observed at 30-60 cm in the sediment (Fauchald and Jumars, 1979). *G. alba* and *S. inflatum* have previously been described as biodiffusors, and *S. inflatum* has also been described as a gallery diffusor (Table 1). No information on the reworking behavior of *L. jeffreysii* was found in the literature. This species was assumed to be functionally similar to *S. inflatum* due to their

close affinity. *S. inflatum* create burrows through crack propagation (Dorgan et al., 2006). They are known as active gallery-creating burrowers that feed on detritus found in the sediment, but can also access surface deposited material (Dorgan et al., 2006;Fauchald and Jumars, 1979). In Gilbert et al. (2007), *S. inflatum* was found to have only limited effect on sediment mixing although luminophores deposited at ~3 cm were transported both down- and upwards in the sediment column. In the current study, activities by *S. inflatum* affected depth variables in all replicates. In

addition, in one aquarium reworking activities also affected the bulk variables which indicate a potential for bulk displacement of sediment. When comparing *L. jeffreysii* with *S. inflatum* , *L. jeffreysii* seemed to be the more effective bioturbator in all measured variables, except for burrow depth. Overall, *S. inflatum* created deeper burrows than *L. jeffreysii* (e.g. Figs. 2, 4 and 5). The difference could in part be explained by the larger size of the *L. jeffreysii* individuals, or suggest a functional difference in the two species.

Reworking activities by *G. alba* were similar to those of *S. inflatum* and *L. jeffreysii* despite a different feeding mode by *G. alba*. These three species created galleries in the sediment with a potential for significant bulk transport (although not always expressed) in combination with fast transport of tracer to deeper sediment layers. While the behavior of these species is not as well-documented as that of the archetypal gallery-diffusor *Hediste diversicolor*, observations in this study supported the classification as gallery-diffusors.

The technique of extracting and processing a range of variables related to particle reworking rather than a few parameters from an a priori chosen model, not only facilitated functional separation between otherwise seemingly similar groups of fauna, but also included mechanisms and functional behavior not directly described by existing models. In the present study, for example, gallery-builders that influenced bulk sediment transport could be separated

from gallery-builders with minor influence on the bulk transport of particles.





## 5 Conclusions

Multivariate analysis based on a suite of reworking variables constitutes a general analytical tool for high-resolution classification of benthic macrofauna according to patterns of particle reworking.

5    The applied variables for particle transport were separated according to two main transport mechanisms, describing the bulk sediment transport from the sediment surface and the depth of particle relocation.

Controls and treatments of macrofauna were separated and grouped according to mode of particle transport. Effects from bioturbation by *N. pernula, A. nitida* and *B. lyrifera* were mainly related to bulk sediment transport while

10    consequences from a transport of particles by *N. incisa* and *T. sarsii* were mainly associated with relocation of particles to deeper sediment layers. *G. alba*, *L. jeffreysii* and *S. inflatum* grouped together with patterns of particle reworking described by both these transport modes.



**Acknowledgements**

This study was financially supported by the Swedish Research Council (VR) and the faculty of sciences, University of Gothenburg. Karl Norling supported during field work and Stefan Agrenius assisted during speciation of benthic macrofauna. Gary Banta kindly provided helpful comments of the manuscript. The support from the crew onboard
5   R/V Oscar von Sydow and staff at the Sven Lovén Center, Kristineberg, is acknowledged.





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





**Figure captions**

Figure 1: Schematic overview of the optical set-up used for high-throughput time-resolved imaging of macrofaunal displacement of fluorescently labeled particles added to the sediment surface. Main components include the LED (570 nm) light source (A) and the CCD-camera (CMOS image sensor) with a bandpass filter (610 nm ± 10nm) mounted on-line for quantification of fluorescence emission (B). The optical components were enclosed in a black box of PVC made in-house. Aquaria were individually positioned in a custom-built high precision rotation stage with a 24 sample cell turret. Image acquisition was controlled from an adjacent laboratory.

Figure 2: Maximum depth of luminophore penetration (MPD) with time during incubations of benthic macrofauna; Nephtys incisa, Abra nitida, Nuculana pernula, Thyasira sarsii, Glycera alba, Lipobranchius jeffreysii, Scalibregma inflatum, Brissopsis lyrifera and the control aquaria. The respective replicates have separate color codes ($n_{faunal\ treatments}$ = 4, $n_{controls}$ = 3). Experiments were performed in 2008 (top row) and 2009 (bottom row). The solid line depicts the maximum depth of animal activity visible from the aquaria wall (Burrow). Maximum sediment depth of the aquaria was 15 cm.

Figure 3: Rate of particle transport (Rate), the biodiffusion coefficient (Db) and the non-local transport coefficient (r). Bioturbation proxies were modeled from the macrofaunal treatments; Nephtys incisa, Abra nitida, Nuculana pernula, Thyasira sarsii, Glycera alba, Lipobranchius jeffreysii, Scalibregma inflatum, Brissopsis lyrifera and the control aquaria. The respective replicates have separate color codes (nfaunal treatments = 4, ncontrols = 3). Experiments were performed in 2008 (left group) and 2009 (right group).

Figure 4: 2D particle redistribution (2Dredist) with time of incubation for the macrofaunal treatments; Nephtys incisa, Abra nitida, Nuculana pernula, Thyasira sarsii, Glycera alba, Lipobranchius jeffreysii, Scalibregma inflatum, Brissopsis lyrifera and the control aquaria. The respective replicates have separate color codes ($n_{faunal\ treatments}$ = 4, $n_{controls}$ = 3). Experiments were performed in 2008 (top row) and 2009 (bottom row).

Figure 5: Daily particle transport for the macrofaunal treatments *Nephtys incisa*, *Abra nitida*, *Nuculana pernula*, *Thyasira sarsii*, *Glycera alba*, *Lipobranchius jeffreysii*, *Scalibregma inflatum*, *Brissopsis lyrifera* and the control aquaria. The respective replicates have separate color codes ($n_{faunal\ treatments}$ = 4, $n_{controls}$ = 3). Experiments were performed in 2008 (top row) and 2009 (bottom row).

Figure 6: Principal component analysis (PCA; Simca v. 14.0, Umetrics AB, Sweden) of the 2008 (top) and 2009 (bottom) experiments. The faunal treatments included *Nephtys incisa* (Ni), *Abra nitida* (An), *Nuculana pernula* (Np), *Thyasira sarsii* (Ts), *Glycera alba* (Ga), *Lipobranchius jeffreysii* (Lj), *Scalibregma inflatum* (Si), *Brissopsis lyrifera* (Bl) and the control aquaria (C). Six proxies for particle reworking (MPD, Burrow, 0.5cm, 2Dredist, Rate, and Rug) were included in the final model ($R^2$ = 0.91 and $Q^2$ = 0.75 for 2008 and $R^2$ = 0.86 and $Q^2$ = 0.60 for 2009). Data is presented by a combined score and loading plot.

Figure 7: Cluster analysis (ngroups = 3) of the 2008 (top) and 2009 (bottom) experiment using data from the PCA-modelling. The faunal treatments included Nephtys incisa (Ni), Abra nitida (An), Nuculana pernula (Np), Thyasira sarsii (Ts), Glycera alba (Ga), Lipobranchius jeffreysii (Lj), Scalibregma inflatum (Si), Brissopsis lyrifera (Bl) and the control aquaria (C).



**Table 1: Functional behavior of the species investigated according to previous studies of mobility, reworking mode and food resource exploited.**

| Species | Feeding mode | Mobility | Reworking behaviour | References |
|---|---|---|---|---|
| | | | | |
| Annelida | | | | |
| *Glycera alba* | Predator | Mobile | Gallery of burrows Biodiffuser | 1,2 |
| *Nephtys incisa* | Predator/Subsurface deposit | Mobile | Transient burrows; Biodiffuser | 1,3,4 |
| *Scalibregma inflatum* *Lipobranchius jeffreysii* | Surface/subsurface deposit | Mobile | Gallery of burrows; Gallery diffuser, Biodiffuser | 1,2,5 |
| | | | | |
| Echinodermata | | | | |
| *Brissopsis lyrifera* | Subsurface deposit | Mobile | Bulldozer; Biodiffuser | 2,6 |
| | | | | |
| Mollusca | | | | |
| *Abra nitida* | Surface deposit/Suspension | Variable | Biodiffuser, Surficial modifier | 2,5,7 |
| *Nuculana pernula* | Subsurface deposit | Semi-mobile | Surficial modifier | 2 |
| *Thyasira sarsii* | Symbiotic | Semi-mobile | Burrows; Downward conveyor | 2,8 |

1. Fauchald and Jumars (1979), 2. Queiros et al. (2013), 3. Sanders (1960), 4. Michaud et al. (2010), 5. Gilbert et al. (2007),
6. Duchene and Rosenberg (2001), 7. Grémare et al. (2004) 8. Dufour and Felbeck (2003). *No information on the functional behavior of *Lipobranchius jeffreysii* and *Nuculana pernula* was found in the literature. These species were assumed to be functionally similar to *Scalibregma inflatum* and *Nuculana minuta*, respectively, due to their close affinity.



**Table 2: Overview of experimental activities.**

| Day 2008 | Day 2009 | Activity |
| --- | --- | --- |
| -54 | -45 | Field sampling of sediment |
| -43 to -46 | -34 to -36 | Aquaria were filled with sediment |
| -43 | -31 | $N_2$ treatment of the overlying water was initiated to remove possible fauna remaining in the sediment by asphyxiation |
| -36 | -27 | Air treatment of the overlying water was initiated to reoxygenate the surface sediment |
| -19 | -17 | Field sampling of benthic macrofauna |
| -12 to -13 | -8 to -9 | Addition of fauna to experimental aquaria |
| 0 | 0 | Addition of luminophores and start of tracer incubations |
| 0 to 14 | 0 to 14 | Sequential capturing of images |
| 14 to 16 | 14 to 15 | Termination of incubations |





**Table 3: Biomass and biovolume of benthic macrofauna. Each aquarium contained one individual, corresponding to an abundance of 460 ind. m$^{-2}$ (n=4).**

| Species | Biomass g aquarium$^{-1}$ (g m$^{-2}$) | | Biovolume mL aquarium$^{-1}$ (mL m$^{-2}$) | |
|---|---|---|---|---|
| *Glycera alba* | 0.16-0.23 | (74-150) | 0.20-0.45 | (92-210) |
| *Lipobranchius jeffreysii* | 0.21-0.46 | (98-210) | 0.30-0.50 | (140-230) |
| *Nephtys incisa* | 0.06-0.14 | (28-65) | 0.10-0.15 | (46-69) |
| *Scalibregma inflatum* | 0.13-0.20 | (62-94) | 0.15-0.30 | (69-140) |
| *Brissopsis lyrifera* | 0.19-0.24 | (88-110) | 0.20-0.30 | (92-140) |
| *Abra nitida* | 0.14-0.23 | (65-110) | 0.20 | (92) |
| *Nuculana pernula* | 0.59-0.64 | (270-300) | 0.40 | (180) |
| *Thyasira sarsii* | 0.056-0.078 | (26-36) | 0.05-0.10 | (23-46) |





**Table 4: Experimental variables that describe particle reworking. Relative amount (%) of luminophores added to the sediment surface that was observed below 0.5, 2 and 4 cm sediment depth (0.5cm, 2cm and 4cm), and the sum of distances between the shallowest and deepest penetration of luminophores observed during image analysis (Rug). Values are given as mean ± standard deviation.**

| Treatment | 0.5cm (%) | 2cm (%) | 4cm (%) | Rug (cm) |
|---|---|---|---|---|
| *Glycera alba* | 4.3 ± 3.5 | 0.4 ± 0.6 | 0.3 ± 0.6 | 930 ± 320 |
| *Lipobranchius jeffreysii* | 9.1 ± 9.3 | 2.9 ± 3.2 | 1.1 ± 1.3 | 1180 ± 980 |
| *Nephtys incisa* | 1.9 ± 2.5 | 0.4 ± 0.4 | 0.2 ± 0.2 | 600 ± 210 |
| *Scalibregma inflatum* | 1.2 ± 2.1 | 0.1 ± 0.2 | 0.0 ± 0.0 | 800 ± 250 |
| *Brissopsis lyrifera* | 4.3 ± 7.9 | 0.1 ± 0.1 | 0.0 ± 0.0 | 650 ± 130 |
| *Abra nitida* | 6.8 ± 9.6 | 0.0 ± 0.0 | 0.0 ± 0.0 | 580 ± 290 |
| *Nuculana pernula* | 24.0 ± 12.2 | 0.0 ± 0.0 | 0.0 ± 0.0 | 910 ± 450 |
| *Thyasira sarsii* | 1.6 ± 3.0 | 0.1 ± 0.0 | 0.1 ± 0.1 | 510 ± 70 |
| Control 2008 | 0.0 ± 0.0 | 0.0 ± 0.0 | 0.0 ± 0.0 | 310 ± 50 |
| Control 2009 | 0.1 ± 0.2 | 0.0 ± 0.0 | 0.0 ± 0.0 | 650 ± 80 |





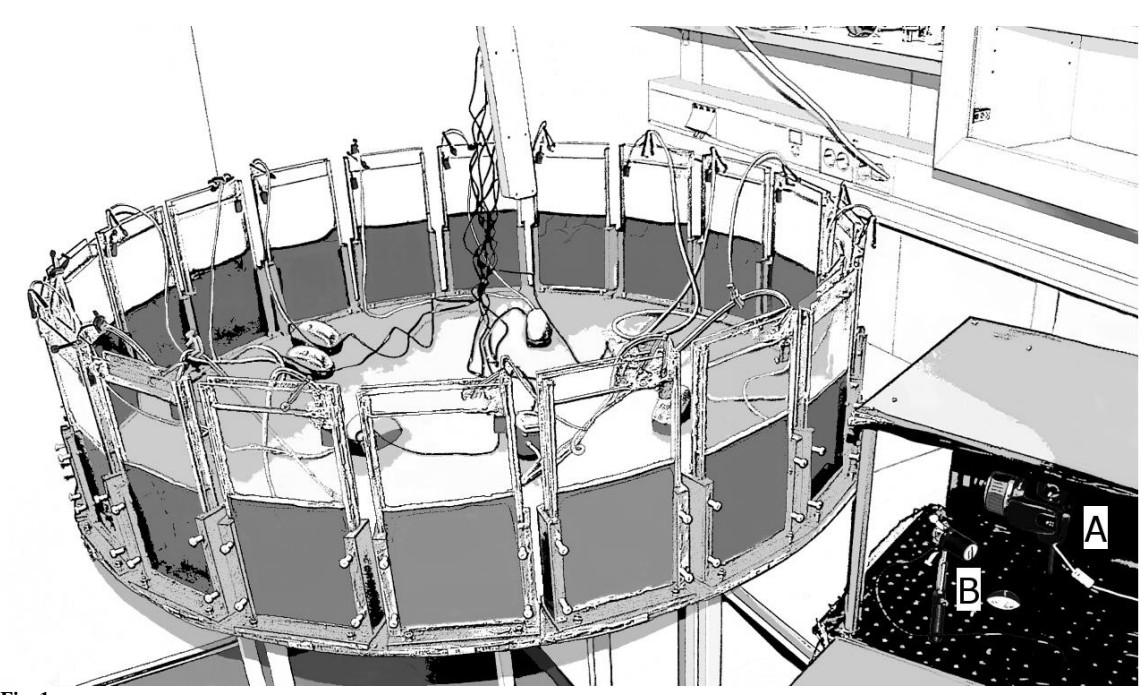

5    **Fig. 1**




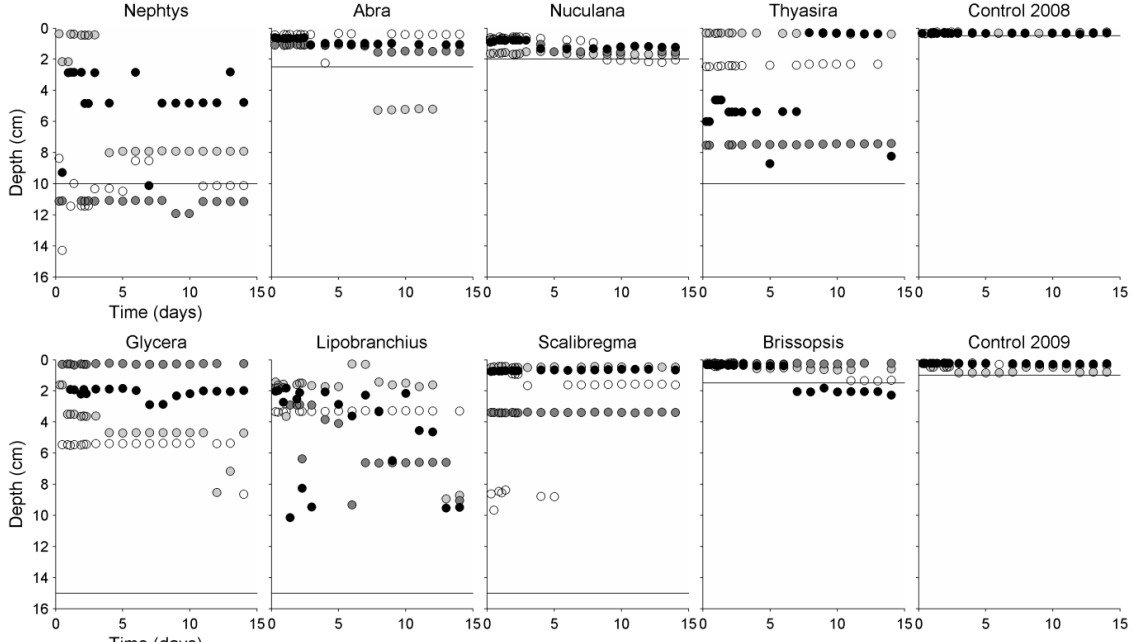

**Fig. 2**





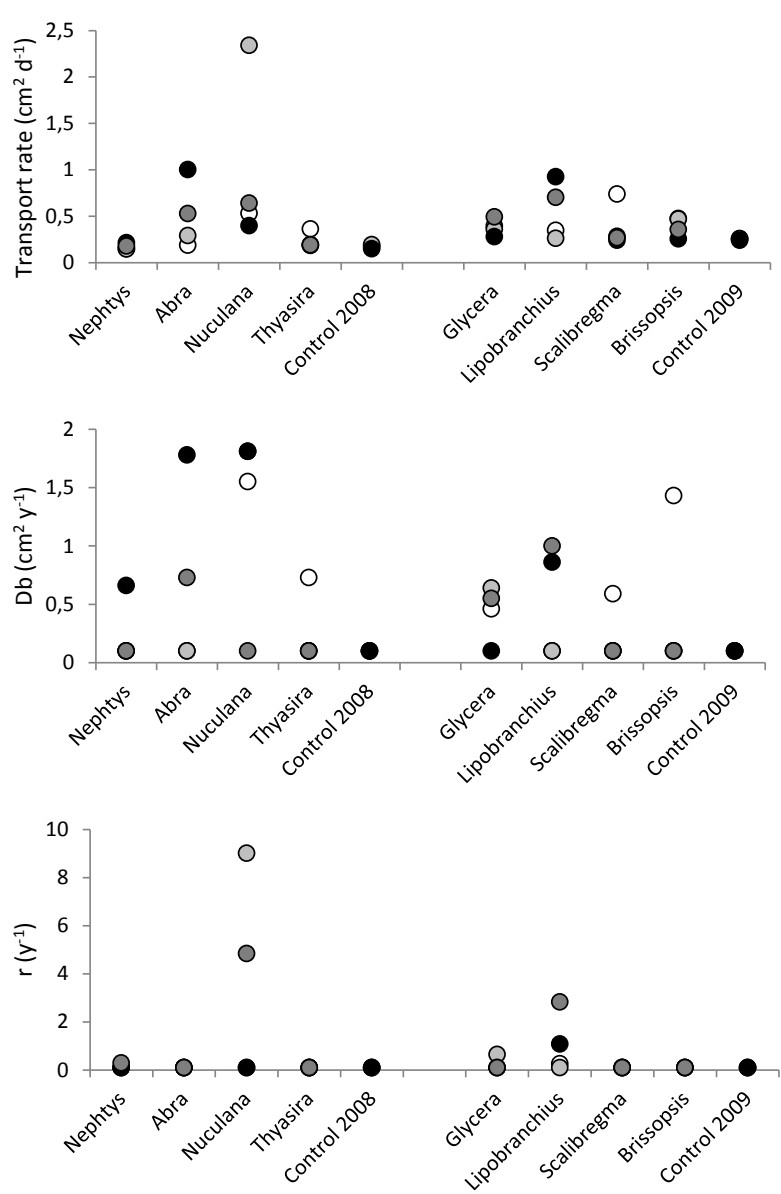

**Fig. 3**





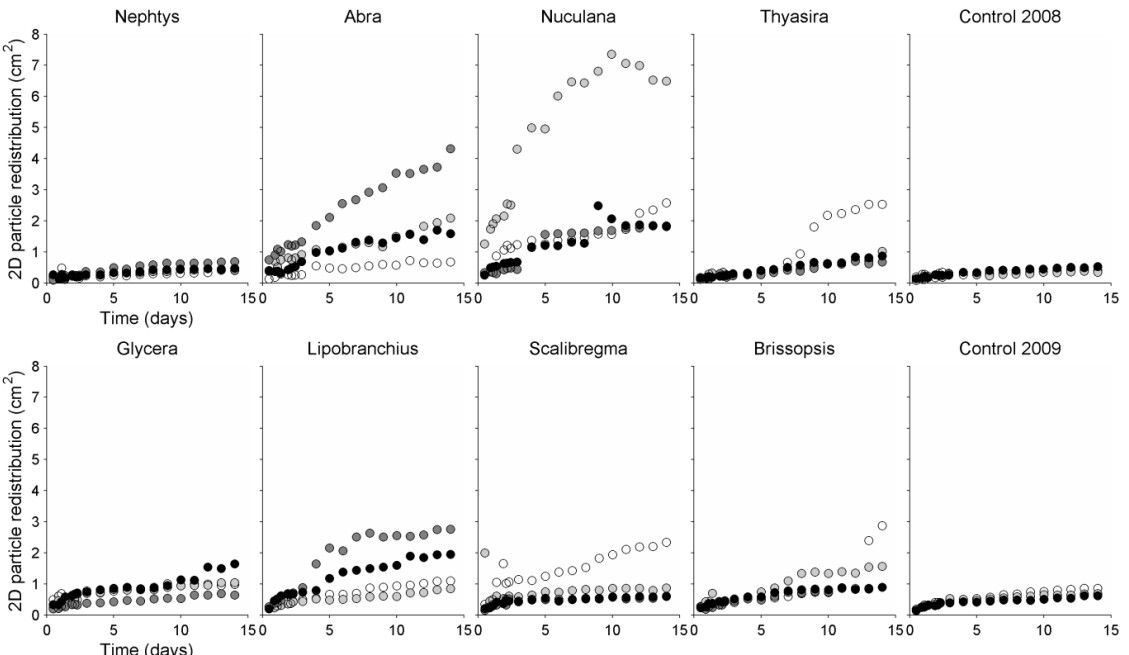

**Fig. 4**

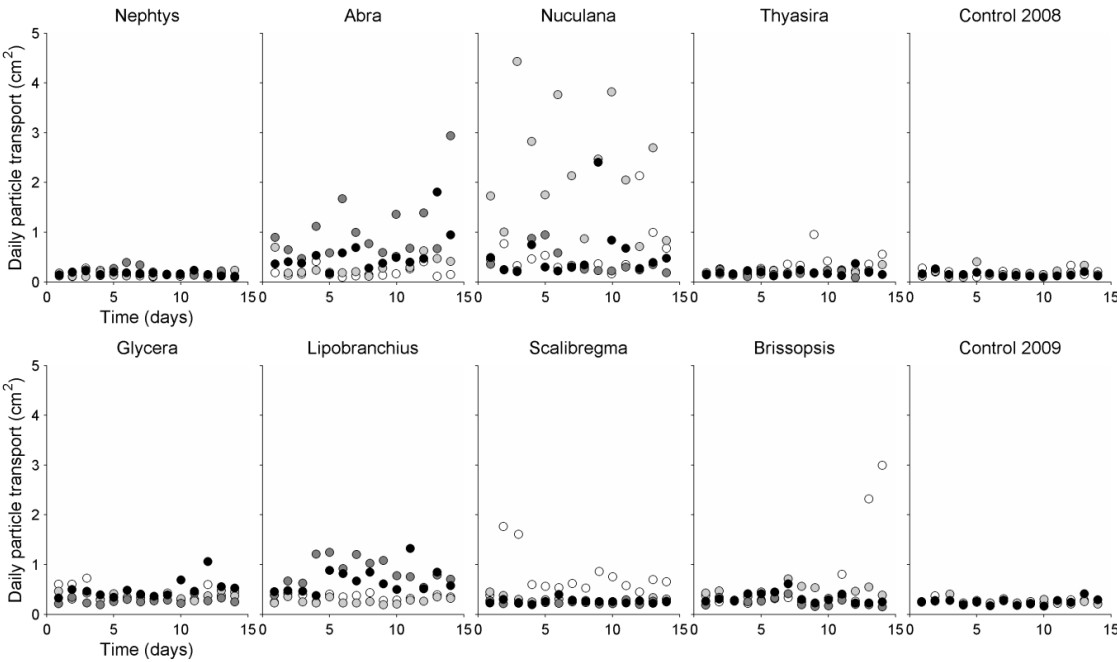

5      **Fig. 5**





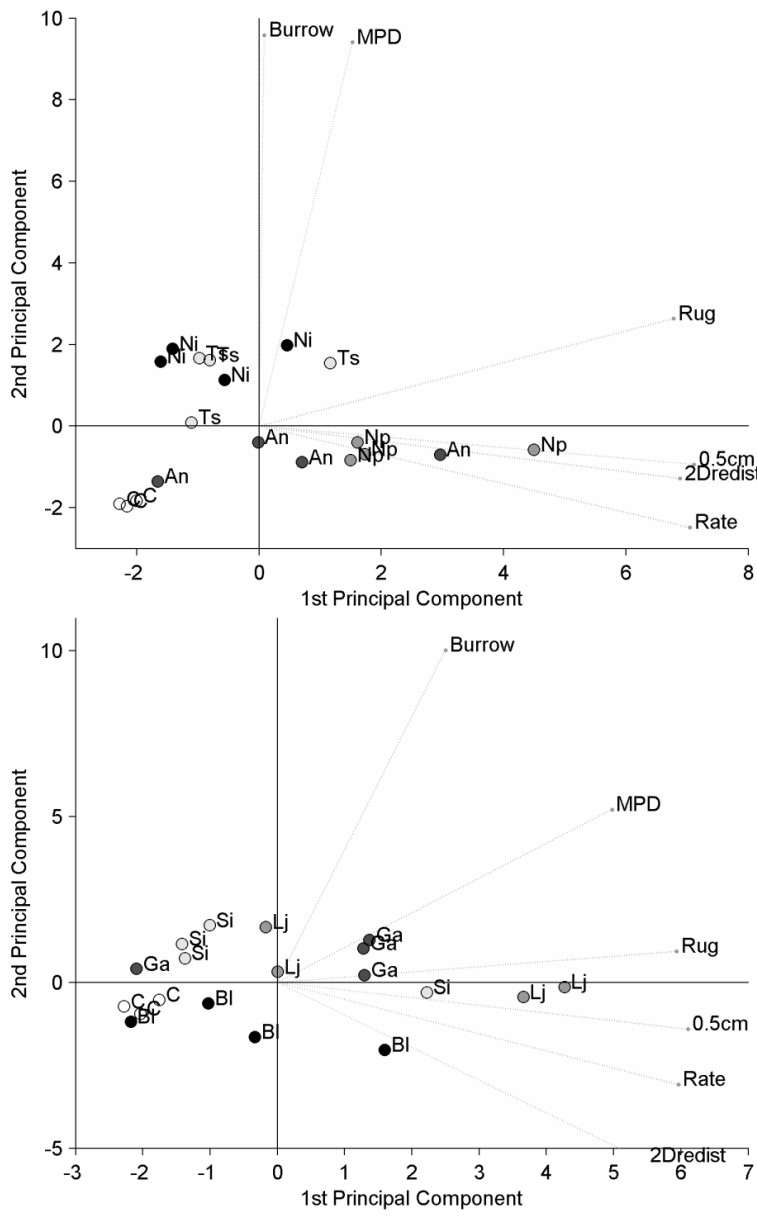

**Fig. 6**




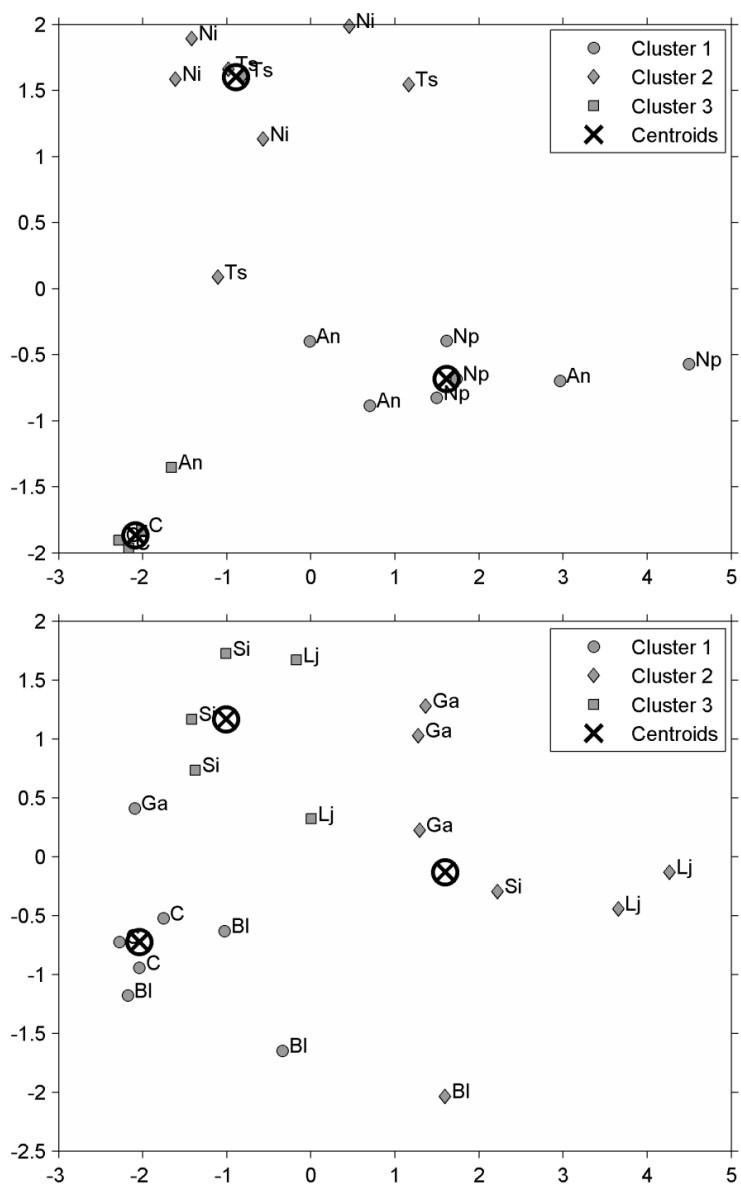

**Fig. 7**