# Peer review of "Functional classification of bioturbating macrofauna in marine sediments using time-resolved imaging of particle displacement and multivariate analysis"

_Biogeosciences, 2016_

## Referee Comment (RC1) · Anonymous Referee #1 · 24 Nov 2016

General comments:

The paper applies a time-resolved imaging method for the functional classification of eight species of bioturbating macrofauna. The text is generally well written and structured, and contains information useful for marine ecologists or biogeochemical modelers. However, in my view, the scope of the paper is rather specialized. As I explain in more detail below, I think the paper has a few deficiencies: 1. the aim of the study as well as the presentation and meaning of the data is not sufficiently clear; 2. the implications of the findings are insufficiently explained in the broader biogeochemical context; 3. the statistical approach used seems to me not fully appropriate as it mixes a subject-independent method (PCA) with a subject-dependent method (choice of specific, possibly not robust, variables *derived* from the raw dataset) to arrive at conclusions.

I fully agree that bioturbating macrofauna play an important role in the biogeochemistry of sediments. Also I can understand that dividing bioturbating macrofauna into functional groups is useful when dealing with complex communities and their impact on sediment biogeochemistry. All this is well explained in the introduction. However, what I do not really understand is the actual *aim* of this study within this context.

If the aim was to classify eight specific species, then the study is rather specialized and perhaps more suitable for a more biological journal. In this context, it is also not clarified why these particular species were selected. If the aim was to demonstrate the utility of the method, then I find the description of the method and the data obtained insufficient to understand (i) what was really done, (ii) why the raw dataset was reduced to the specific variables, and (iii) what the differences between the studied species really mean.

Personally, I prefer to look at raw data first and only then explore how they were processed, analyzed and interpreted by others. However, this study does not make it possible to do this as no measured data is presented. What is presented are only variables *derived* from the raw data based on a specific choice of the authors. This specific choice may be biased towards the aim, possibly influencing data interpretation. However, there is no way to find out.

These days there is plenty of room for including images or videos as supplementary information, and I encourage the authors to do this. Specifically, it would be really useful if they present their raw data as videos showing frames with the luminophores distributions as a function of time. Application of the threshold would be fine, and the replicate images for each species could be combined into one frame. They don't have

to be in the full resolution, but at least an impression of "how the distributions looked like" would be useful. Then we can understand the complexity of the dataset as well as fully appreciate the need to reduce it. Additionally, we can better judge whether the approach to reduce the dataset, as chosen by the authors, was appropriate.

In this context, I have some reservations about the statistical analysis employed to arrive at conclusions. It is based on PCA of a set of variables derived from the raw data. However, I am curious why PCA was not used on the raw data itself. The whole point of PCA is to reduce complexity of the dataset about a system into variables "that matter" most in its description. However, a priori reduction of the raw data into derived variables risks that this analysis is biased, because it gives higher statistical weight to some data (in the raw dataset) compared to others. Therefore, I think the authors need to dedicate some space justifying their statistical analysis approach. In my view, the classification of sediment reworking types/modes that the authors present should *emerge* from the PCA analysis of the raw data. Instead, the authors *bundle* the raw data in such a way that the sought after classification of a specific species is either accepted (with some caveats) or rejected.

I am particularly concerned about the maximum penetration depth (MPD) variable. As I understand it from the text, this variable is essentially a measure of rare events. That is, if a particle from the surface is by chance deposited at some depth in the sediment, it can remain there for a long time. This will lead to a constant value of MPD over extended periods of time (as indeed shown in Fig. 2), giving this one (likely rare) event a disproportionately large weight in the final PCA analysis of the (reduced) data.

Also, it is not entirely clear how certain the estimates of the parameters shown in Fig. 3 are. One can always fit a dataset with a model and obtain a process parameter such as r or Db. But the authors do not provide any clues as to the quality of their data fits and uncertainties of the estimated parameters. It would help if a few examples (best and, possibly, worse) of the data together with the fits are shown and the above points are discussed.

Last but not least, the quantities characterizing the 2D redistribution of the particles shown in Fig. 5 are unclear. As I understand it, redistribution of particles is a 2D image. But how is it reduced to a number in cm^2? Obviously, there is an enormous data reduction involved. If there is no other metric used to characterize it (e.g., some sort of variance), then again this data points will have disproportionately large weight in the final PCA analysis.

Specific comments/questions:

p.3

l.16: you make a distinction between functional groups and functional modes. In the introduction these subtleties are not clearly explained.

l.21: After reading the introduction, I do not understand the aim of the study, why the specified animals were chosen, and how this choice was related to the classification mentioned in the preceding text.

l.25: what's the point of (n=4) here?

l.32: what do you mean by "gallery-diffusor model could be *evaluated*"?

l.35: I am quite confused about terminology used with respect to functional group vs. behavior, or reworking mode vs. behavior.

p.5

l.29: unclear formulation: for each column? or for two luminophore particles?

l.31: incorrect use of "summarized". perhaps "summed"?

l.35: No results of this evaluation shown. Hard to get a feeling for it, and numbers don't tell much.

p.7

l.15-25: The results are presented in a very inacurate way, making it hard to decide

how to understand them. Check out the following words that are scattered throughout the paragraph and be more specific about their meaning: occasionally, a significant downward transport, sporadically observed, suggested, specific sampling occasions, mainly observed, clear tendency of similar patterns.

l.26: why do you exclude G. alba and B.lyrifera from this list? Their values are also quite large, and, looking at the STD, probablz not significantlz different from the other three mentioned species.

l.34: what do you mean by indications?

Table 1: since the values do not add up to 100%, I wonder where the rest of the luminophores goes. What about the wall effect, i.e., the transport of luminophores that are transported away from the wall and are thus invisible from the side?

p.8

l.1: Unclear what the "transport rate" shown in Fig. 3 really means. Also the meaning of the unit is unclear.

l.4: Unclear what you really show in Fig. 4. According to methods, 2D redistribution reflects Mt-M0, which is a matrix. But all I see in Fig. 4 points. It is unclear what these points have to do with the 2D character of this metric.

l.5-7: Also these sentences do not make things much clearer.

l.8: I see this only for Glycera and Scalibregma, which is not "in general".

l.15: Why did this occur?

l.21: So this is the measure of bulk sediment transport? Please clarify how/why.

p.10

l.5: Could you not look at this aspect (frequency distributions) in your data? Or is this where the Db and r come from? Not clear how you got them.

l.17-19: unclear what the difference is between diffusive-like mixing mode and diffusive-like transport over long time scales.

p.13

l.21-24: Nice, but you still made an a priori choice of a model when you were choosing the variables derived from your raw data.

p.21

Table 1: Unclear meaning of terminology. What does it mean "Feeding mode = Subsurface deposit"? And if you say "variable mobility", variable between what? Or "semi-mobile" - what exactly is that?

---

## Author Comment (AC1) · 22 Dec 2016

Response from the authors to the interactive comment on the manuscript "Functional classification of bioturbating macrofauna in marine sediments using time-resolved imaging of particle displacement and multivariate analysis" (#bg-2016-411) by S. Lindqvist et al. for publication in Biogeosciences.

Summary of the response

[Figure]

We thank the reviewer for the thorough comments that significantly helped to improve the revised version of our manuscript. We are grateful for the detailed insight and acknowledge that some information appeared implicit and needed further clarifications. In accordance with comments provided by the reviewer and to improve the transparency of the raw data quality as well as to clarify arguments included in the manuscript, supplementary material has been included. This material includes videos with time-laps sequences of luminophore distributions and figures that illustrate how well experimental results fit to the gallery diffusion model. We have in the revised version also tried our best to further clarify our aims for the study. Further, variables used during the multivariate evaluation have been described in more detail and biogeochemical consequences from our study have been addressed in the discussion of the revised version of the manuscript. We have also emphasized to clarify the usefulness of the applied method in a broader context as the overriding principles are general and can be used in a wide range of analytical and methodological settings.

We have to the best of our ability tried to meet the comments provided by the reviewer and we now hope that our manuscript will be accepted for publication as a regular article in Biogeosciences.

Yours sincerely

Stina Lindqvist /on behalf of all authors

  General comments:

Reviewer: . . . However, in my view, the scope of the paper is rather specialized. As I explain in more detail below, I think the paper has a few deficiencies: 1. the aim of the study as well as the presentation and meaning of the data is not sufficiently clear; 2. The implications of the findings are insufficiently explained in the broader biogeochemical context; 3. the statistical approach used seems to me not fully appropriate as it mixes a subject-independent method (PCA) with a subject-dependent method (choice of specific, possibly not robust, variables *derived* from the raw dataset) to arrive at

conclusions.

Response: These three main concerns are addressed in detail under the bolded headlines below.

Reviewer: I fully agree that bioturbating macrofauna play an important role in the biogeochemistry of sediments. Also I can understand that dividing bioturbating macrofauna into functional groups is useful when dealing with complex communities and their impact on sediment biogeochemistry. All this is well explained in the introduction. However, what I do not really understand is the actual *aim* of this study within this context. If the aim was to classify eight specific species, then the study is rather specialized and perhaps more suitable for a more biological journal. In this context, it is also not clarified why these particular species were selected. If the aim was to demonstrate the utility of the method, then I find the description of the method and the data obtained insufficient to understand (i) what was really done, (ii) why the raw dataset was reduced to the specific variables, and (iii) what the differences between the studied species really mean.

Response: These comments are addressed under the Aim and Method-headlines below.

Reviewer: Personally, I prefer to look at raw data first and only then explore how they were processed, analyzed and interpreted by others. However, this study does not make it possible to do this as no measured data is presented. What is presented are only variables *derived* from the raw data based on a specific choice of the authors. This specific choice may be biased towards the aim, possibly influencing data interpretation. However, there is no way to find out. These days there is plenty of room for including images or videos as supplementary information, and I encourage the authors to do this. Specifically, it would be really useful if they present their raw data as videos showing frames with the luminophores distributions as a function of time. Application of the threshold would be fine, and the replicate images for each species

could be combined into one frame. They don't have to be in the full resolution, but at least an impression of "how the distributions looked like" would be useful. Then we can understand the complexity of the dataset as well as fully appreciate the need to reduce it. Additionally, we can better judge whether the approach to reduce the dataset, as chosen by the authors, was appropriate.

Response: We agree to the overriding principles that raw data, whenever possible, should be included along with more detailed calculations and modelling of the data. As suggested by the reviewer we have as supplementary information along with this response to the reviewer's comments enclosed time-frame videos of raw data that illustrate the distribution of luminophores as a function of time for the various treatments. Videos were made up from separate images of luminophore distributions.

Response: Comments related to the aim of the study Our main objective was indeed to investigate functionality of benthic macrofauna in a broad perspective and to describe the utility of the multivariate approach using a wide range of variables that describe a function essential for the system (particle transport). It was not, however, a primary and a priori goal to classify these specific species in accordance to the obtained variables by multi variate modelling. As pointed out by the reviewer, this distinction may not have been clearly stated in the manuscript. The revised version of the manuscript has been changed in accordance to this general intention of clarification (revised version p.3 l.14-21).

In the original manuscript, section 4.2 was included in part to compare this new approach of classification of fauna according to experimentally derived variables for functionality (particle transport) with previous classifications of these (or similar) species from a quantitative and mechanistic perspective. Overall, there is a general deficiency of studies with detailed investigations of particle reworking by single- or multi-species communities of benthic macrofauna. In response to the reviewer's comments and to clarify the distinction discussed above, we have shorted this section of the manuscript that describes the species of macrofauna used. Specific details of species behaviour

have been removed (revised version p.11-13).

The motivation for selecting each of the eight species of macrofauna was already included in the original manuscript (p.4, l.1-3). We believe these grounds should be considered sufficient, particularly as the overall aim was to experimentally investigate functionality of benthic macrofauna in a broad perspective. No changes made in the revised version of the manuscript.

Response: Comments related to biogeochemical consequences Although quantification of biogeochemical parameters and detailed investigations on the effects of particle transport by various species of macrofauna for element cycling were considered outside the scope of this study, we agree that a section that describes this aspect would further emphasize the generally broad approach. Text in accordance to this aspect was therefore added to the discussion, section 4.2, in the revised version.

Response: Comments related to the method

Reviewer: . . .(i). What was really done. (ii). Why the raw dataset was reduced to the specific variables

. . .What is presented are only variables *derived* from the raw data based on a specific choice of the authors. This specific choice may be biased towards the aim, possibly influencing data interpretation.

. . . However, I am curious why PCA was not used on the raw data itself. The whole point of PCA is to reduce complexity of the dataset about a system into variables "that matter" most in its description. However, a priori reduction of the raw data into derived variables risks that this analysis is biased, because it gives higher statistical weight to some data (in the raw dataset) compared to others. Therefore, I think the authors need to dedicate some space justifying their statistical analysis approach. In my view, the classification of sediment reworking types/modes that the authors present should *emerge* from the PCA analysis of the raw data. Instead, the authors *bundle* the

raw data in such a way that the sought after classification of a specific species is either accepted (with some caveats) or rejected.

...the statistical approach used seems to me not fully appropriate as it mixes a subject-independent method (PCA) with a subject-dependent method (choice of specific, possibly not robust, variables *derived* from the raw dataset) to arrive at conclusions.

Response: In accordance with the reviewer's comments, the methodology and general principles of PCA has now been described in more detail (section 2.5.) to clarify and justify the statistical approach (revised version p.6 l.12-22). Further, we have explained the calculation of the 2D reworking variables in more detail (revised version p.5 l.19-31).

The PCA approach was applied as a means to evaluate the broad selection of variables in a general sense and to facilitate the description of a phenomenon (i.e. particle transport induced by benthic macrofauna), not to simplify the set of raw data. Rather than reducing the binarized images into principal components, we chose to reduce a larger number of reworking variables derived from the images into principal components. Variables used in the final PCA model were chosen based on increasing R2 and Q2 in the PCA. As was also pointed out by the reviewer, experimental variables that describe particle reworking were selected in a rather subjective manner. However, as a general principle, the selection was based on i) previous studies by others (see references in the manuscript), ii) some 20 years of experience working with various aspects of bioturbation by benthic macrofauna, iii) relevance and coupling to biogeochemical processes, and (iv) a judgement that the selection of variables could be extracted from the captured images. Which properties that are actually quantified and later also used during modelling are always up for a subjective decision in some sense.

Using the binary images directly would in our opinion not be an effective means to meet the aim of this manuscript. There would be too much data overshadowing transport processes of significance for biogeochemical processes in bioturbated deposits. We find it more relevant to identify variables directly coupled to transport processes

that have previously been described, for example, events deep in the sediment, events occurring over time and events related to the magnitude of sediment that is displaced by macrofaunal activities. By using variables that previously have been described as important for studies of particle reworking, there is an opportunity to evaluate the variables in accordance with the present experimental settings and compare observations with previous research.

For a general discussion of available techniques to quantify particle reworking, please see e.g. Maire et al. (2008) and references therein. The variables used to describe particle transport in the present study have previously been used by us (e.g. Gilbert et al., 2003; Lindqvist et al., 2013) and by others (e.g. Hedman et al., 2011; Maire et al., 2006; Murray et al., 2014) to quantify reworking by benthic macrofauna.

In the iterative process of finding optimal variables for the data set, the variables were found to consistently group according to their representation of "bulk" or "depth" transport of particles. Additionally, the treatments of benthic macrofauna clustered similarly during the initial screening process as in the final model, which indicated robustness in the model. One of the points made in our manuscript is that reworking variables can be grouped according to whether they quantify sediment reworking/transport in terms of "bulk" (quantity) or "depth" (distance). Such quite simple distinction enables a general classification of benthic macrofauna according to these mechanisms of transport in almost any type of experimental set-up designed to quantify particle reworking.

Reviewer: I am particularly concerned about the maximum penetration depth (MPD) variable. As I understand it from the text, this variable is essentially a measure of rare events. That is, if a particle from the surface is by chance deposited at some depth in the sediment, it can remain there for a long time. This will lead to a constant value of MPD over extended periods of time (as indeed shown in Fig. 2), giving this one (likely rare) event a disproportionately large weight in the final PCA analysis of the (reduced) data.

Response: While vertical events of particle transport to deeper sediment layers may be less frequent and less intense for most species than events of horizontal particle transport at the surface, they are not necessarily less important from a biogeochemical perspective. Patterns of vertical transport of both particles and solutes may not at all be stochastic as some species actually depend on such cross-boundary pathways for their metabolism (e.g. Thyasira sarsii). There are also species that create semi-permanent or permanent burrows into which organic material is transported for feeding. The burrow depth and maximum depth of particle reworking have implications for reaction pathways during element cycling (e.g. for redox reactions) because of the supply of oxidants (either dissolved, e.g. $O_2$ and $NO_3^-$, through ventilation of pore water, or particulate material, e.g. $MnO_2$ and $FeOOH$, through particle relocation) and surficial organic material that would not have been transported to these often anoxic sediment layers in the absence of bioturbation. There is also a potential for inverse transport of reduced particulate material (e.g. feaces) defecated at depth upwards to oxic surficial layers. The use of depth variables that describe temporal and spatial characteristics of vertical particle transport is quite common and often necessary within this line of research.

Reviewer: Also, it is not entirely clear how certain the estimates of the parameters shown in Fig.3 are. One can always fit a dataset with a model and obtain a process parameter such as $r$ or $Db$. But the authors do not provide any clues as to the quality of their data fits and uncertainties of the estimated parameters. It would help if a few examples (best and, possibly, worse) of the data together with the fits are shown and the above points are discussed.

Response: To illustrate the coupling between raw data and results from the gallery diffusor model, we have included figures of luminophore distributions with depth (2-D) along with model fit describing the specific distributions as supplementary material. Additionally, model fits were presented in section 3.3 (p.8 l.6-7) in the revised version of the manuscript.

[Figure]

Reviewer: Last but not least, the quantities characterizing the 2D redistribution of the particles shown in Fig. 5 are unclear. As I understand it, redistribution of particles is a 2D image. But how is it reduced to a number in cmËĘ2? Obviously, there is an enormous data reduction involved. If there is no other metric used to characterize it (e.g., some sort of variance), then again this data points will have disproportionately large weight in the final PCA analysis.

Response: The 2D reworking variables (i) 2D redistribution (ii) daily transport and (iii) transport rate were calculated from the time series of images. They represent a measure of 2-D transport but are not in 2D as such. The manuscript was revised to further clarify how variables for 2-D redistribution were estimated from images of luminophore redistribution in section 2.4 (revised version p.5 l.19-29)

All replicate data points are plotted in Fig. 4 and 5 to facilitate transparency in the quality of the dataset.

Reviewer: Specific comments/questions: p.3 l.16: you make a distinction between functional groups and functional modes. In the introduction these subtleties are not clearly explained.

Response: Functional mode is how a functional outcome or performance is achieved. Functions take place in a mode at a performance level (intensity) by a portion of the members of a species (could be different at different life stages). A functional group is a group of organisms that perform similar functional modes that make it possible to classify them into a fairly coherent group of bioturbators that have a similar effect on the sediment transport properties. A difference in functional mode of particle reworking would facilitate grouping into functional groups. Based on the comment by the reviewer, we have in this context replaced functional group with functional mode in the revised version, p.2 l.19.

Reviewer: l.21: After reading the introduction, I do not understand the aim of the study, why the specified animals were chosen, and how this choice was related to the classification mentioned in the preceding text.

Response: Response to this comment on the aim of the study is provided above and revisions have been made in the revised manuscript.

Reviewer: l.25: what's the point of (n=4) here?

Response: This refers to the number of replicates for each treatment i.e. number of aquaria. 10 treatments were examined and they were replicated 4 times = 40 aquaria. The sentence was slightly modified to clarify (revised version p.3 l.24).

Reviewer: l.32: what do you mean by "gallery-diffusor model could be *evaluated*"?

Response: Typo. The model was not evaluated. The sentence has been revised to clarify (revised version p.3 l.32).

Reviewer: l.35: I am quite confused about terminology used with respect to functional group vs. behavior, or reworking mode vs. behavior.

Response: As mentioned above, the text has been revised to clarify where appropriate.

Reviewer: p.5 l.29: unclear formulation: for each column? or for two luminophore particles?

Response: The description of the variable rugosity has been revised to further clarify that distances were calculated for each column and then summed (revised version p.5 l.30-32).

Reviewer: l.31: incorrect use of "summarized". perhaps "summed"?

Response: We agree. Summarized has been changed to summed (revised version p.5 l.33).

Reviewer: l.35: No results of this evaluation shown. Hard to get a feeling for it, and numbers don't tell much.

Response: As mentioned above, we have included figures on luminophore distributions

along with modeled data and error values as supplementary material. By providing this additional information the coupling between presented data and results from the model description can be further evaluated.

Reviewer: I p.7 l.15-25: The results are presented in a very inaccurate way, making it hard to decide how to understand them. Check out the following words that are scattered throughout the paragraph and be more specific about their meaning: occasionally, a significant downward transport, sporadically observed, suggested, specific sampling occasions, mainly observed, clear tendency of similar patterns.

Response: To meet this comment we have tried to be more specific and stringently when describing observations. Descriptions of individual aquaria have been removed from the text with more focus on the general trends. Revisions were made in section 3.2, revised version p.7 l.21-32.

Reviewer: l.26: why do you exclude G. alba and B.lyrifera from this list? Their values are also quite large, and, looking at the STD, probablz not significantlz different from the other three mentioned species.

Response: Results for individual species are summarized in Table 4. In accordance with the ambition to reduce the focus on observations for individual species, we have tried to highlight species that were particularly important for the respective variable. The impact from activities related to G. alba and B.lyrifera was considered as "intermediate" for the variable 0.5 cm in the comparison between species. No changes made.

Reviewer: l.34: what do you mean by indications?

Response: There were somewhat contradictory observations from the distribution of luminophores (which affected e.g. the variable MPD) in the N. pernula treatmentand the non-local coefficients described by r in the model fit.

Reviewer: Table 1: since the values do not add up to 100%, I wonder where the rest of the luminophores goes. What about the wall effect, i.e., the transport of luminophores

that are transported away from the wall and are thus invisible from the side?

Response: It is not clear if this comment refers to Table 1 or Table 4. We believe the reviewer refers to Table 4 rather than Table 1. The numbers are not meant to add up to 100% as they describe the relative amount (%) of total luminophores observed in the image that were found beneath a certain depth (0.5, 2 and 4 cm). Examples of luminophore distributions along with model fits are included as supplementary material in the revised version of the manuscript. No additional revisions were made.

Reviewer: p.8 l.1: Unclear what the "transport rate" shown in Fig. 3 really means. Also the meaning of the unit is unclear.

Response: As mentioned above, the description in the method section has been revised to further clarify transport rate and the unit used.

Reviewer: l.4: Unclear what you really show in Fig. 4. According to methods, 2D redistribution reflects Mt-M0, which is a matrix. But all I see in Fig. 4 points. It is unclear what these points have to do with the 2D character of this metric.

Response: Please see the response above. Revisions were made to clarify that the variable is derived from the 2D images, not a matrix itself. Variables derived from the luminophore distributions (1D) are also numbers – not vectors (e.g. MPD, Db, r, 0.5 cm).

Reviewer: l.8: I see this only for Glycera and Scalibregma, which is not "in general".

Response: We disagree. In the plot where particle redistributions are illustrated vs time (Fig. 4), the slope was steepest during the first couple of days for most aquaria. No changes made.

Reviewer: l.21: So this is the measure of bulk sediment transport? Please clarify how/why.

Response: The 2D redistribution and the daily transport describe the number of pixels

that represent particle transport with time. Transport rate is the sum of the daily transport of particles throughout the experimental period divided by time. Because "bulk transport" was first introduced in section 3.5, "bulk" was removed from section 3.4 in the revised version of the manuscript.

Reviewer: p.10 l.5: Could you not look at this aspect (frequency distributions) in your data? Or is this where the Db and r come from? Not clear how you got them.

Response: In order to evaluate fingerprints for particle mixing, particles must be tracked individually. Such approach would require high-frequency image acquisition (0.1 Hz was used in Bernard et al., 2012 and Bernard et al., 2016). Our experimental setup was focused on longer duration of particle reworking activities (days) and the frequency of image acquisition was therefore lower.

Reviewer: l.17-19: unclear what the difference is between diffusive-like mixing mode and diffusivelike transport over long time scales.

Response: Not clear what the reviewer mean.

Reviewer: p.13 l.21-24: Nice, but you still made an a priori choice of a model when you were choosing the variables derived from your raw data.

Response: See detailed discussion above under Method-headline

Reviewer: p.21 Table 1: Unclear meaning of terminology. What does it mean "Feeding mode = Subsurface deposit"? And if you say "variable mobility", variable between what? Or "semimobile" - what exactly is that?

Response: Occasionally, it is useful to distinguish between a quantitative measure of a function/mode and a qualitative description of the function/mode itself. The mode of mobility and its quantitative measure was originally extracted from different references. For this reason, the description of e.g. mobility was not stringent. To meet the comment provided by the reviewer, descriptions of mobility of fauna were supplemented with information from Queirós et al. (2013) (Table 1; revised version).

Additional comments from the authors:

During production of the supplementary material, an error was observed in Fig. 4. One data point in the Scalibregma inflatum treatment at t=12 h was erroneous. The 2D redistribution was recalculated and Fig. 4 corrected accordingly in the revised version of the manuscript.

References referred to in the response:

Bernard G., Duchene, J. C., Romero-Ramirez, A., Lecroart, P., Maire, O., Ciutat, A., Deflandre, B., Gremare, A: Experimental Assessment of the Effects of Temperature and Food Availability on Particle Mixing by the Bivalve Abra alba Using New Image Analysis Techniques, PLoS ONE, 11, 4, 2016

Gilbert, F., Hulth, S., Strömberg, N., Ringdahl, K., and Poggiale, J. C.: 2-D optical quantification of particle reworking activities in marine surface sediments, Journal of Experimental Marine Biology and Ecology, 285, 251-263, 2003.

Hedman, J. E., Gunnarsson, J. S., Samuelsson, G., and Gilbert, F.: Particle reworking and solute transport by the sediment-living polychaetes Marenzelleria neglecta and Hediste diversicolor, Journal of Experimental Marine Biology and Ecology, 407, 294-301, 10.1016/j.jembe.2011.06.026, 2011.

Lindqvist, S., Gilbert, F., Eriksson, S. P., and Hulth, S.: Activities by Hediste diversicolor under different light regimes: Experimental quantification of particle reworking using time-resolved imaging, Journal of Experimental Marine Biology and Ecology, 448, 240-249, 10.1016/j.jembe.2013.06.014, 2013.

Maire, O., Duchene, J. C., Rosenberg, R., de Mendonca, J. B., and Gremare, A.: Effects of food availability on sediment reworking in Abra ovata and A. nitida, Mar. Ecol.-Prog. Ser., 319, 135-153, 2006.

Maire, O., Lecroart, P., Meysman, F., Rosenberg, R., Duchene, J. C., and Gremare, A.: Quantification of sediment reworking rates in bioturbation research: a review. Aquatic

Biology, 2, 219-238, 2008.

Murray, F., Douglas, A., and Solan, M.: Species that share traits do not necessarily form distinct and universally applicable functional effect groups, Marine Ecology Progress Series, 516, 23-34, 10.3354/meps11020, 2014.

Queirós, A. M., Birchenough, S. N. R., Bremner, J., Godbold, J. A., Parker, R. E., Romero-Ramirez, A., Reiss, H., Solan, M., Somerfield, P. J., Van Colen, C., Van Hoey, G., and Widdicombe, S.: A bioturbation classification of European marine infaunal invertebrates, Ecol. Evol., 3, 3958-3985, 10.1002/ece3.769, 2013.

Please also note the supplement to this comment:
http://www.biogeosciences-discuss.net/bg-2016-411/bg-2016-411-AC1-supplement.pdf

[Figure]

**Supplement:**

[revised manuscript text omitted]
 particles compared to the initial location of luminophores. Only the last value in the time series for 2D redistribution was used in the statistical analysis. The daily transport was calculated by subtracting the binary image obtained at time t ($M_t$), with the binary image obtained the previous day ($M_{t-24 h}$) and summing the absolute values of the resulting matrix. 2D redistribution and daily transport express the number of pixels that represent particle transport. The number of pixels was converted to area units (cm$^2$) for purpose of comparison. The transport rate of particles, slightly modified from that described in Lindqvist et al. (2013), was calculated from the sum of the daily transport ($\Sigma T_{daily}$) divided by the number of experimental days ($\Sigma T_{daily}/14$).

The 2D variable rugosity (Rug; Murray et al., 2014) was extracted from the last image in the time series. The distance between the shallowest and the deepest luminophore was calculated for each column in the image. Rugosity was calculated as the sum of these distances (Murray et al., 2014).

The pixels of vertical layers including luminophores were summed and assigned to the midpoint of each layer to reduce the 2D image to a 1D vertical profile. From these profiles, variables describing the relative fraction (%) of luminophores transported beneath 0.5 (0.5cm), 2 (2cm) or 4 cm (4cm) were extracted. 1D tracer distributions were also used to quantitatively evaluate the particle transport by the gallery-diffusion model, which approximates the

diffusive mixing by the biodiffusion coefficient (Db) and the rapid transport over long distances by the non-local transport coefficient, r (François et al., 2002). The model is time-and space dependent and employs ordinary differential equations to minimize a weighted sum of squared difference between the observed and modeled tracer concentrations with sediment depth.

5    The maximum penetration depth of luminophores (MPD) was calculated from the deepest pixel row containing luminophores (Maire et al., 2006). To exclude noise, the limit of quantification was set to 3 pixels in 2008 and 8 pixels in 2009. The limit was determined from the camera response in deep sediment layers of the control cores with no luminophores (i.e. background). The sediment depth at which animal activity was observed from photos of the aquaria walls taken under normal light conditions was also determined (Burrow).

10    **2.5 Statistical analysis**

Principal component analyses (PCA) were performed in Simca (v. 14.0, Umetrics AB, Sweden) to identify groups with similar particle reworking behavior and to evaluate the most relevant reworking variables to be included in models of functional classification. PCA is a multivariate projection method designed to reduce dimensionality and to identify patterns in a data set. Variables are converted to principal components that form an orthogonal basis for

15    the space of the data. Each principal component is a linear combination of the original variables orthogonal to the preceding component. Prior to analysis, variables were assessed by graphical exploration of box plots (Quinn and Keough, 2002). Where appropriate, variables were log(x) or log(x+1) transformed in order to fit the requirements for PCA. The variables were mean centered and scaled to unit-variance. An initial evaluation showed different behavior for the 2009 experiment compared to 2008, and the two sets were therefore modeled separately. The models were

20    diagnosed by $R^2$ and $Q^2$, representing their ability to describe ($R^2$) and predict ($Q^2$) the variation in the data (Eriksson et al., 2006). A range of approximately a dozen reworking variables derived from images of luminophore distributions were reduced to seven variables included in the final models. These variables best described the variance and predictability of the model data set. A cluster analysis ($n_{groups}$=3) was performed on the results from the PCA in Matlab. Centroids were determined by a two-phase iterative algorithm that minimized the sum of point-to-

[revised manuscript text omitted]

25   local transport was modelled following *A. nitida* reworking activities and observations of subsurface peaks in tracer distributions were found in Maire et al (2006). Although there are indications of non-diffusive transport and potentially deep burial of tracers by *A. nitida*, observations from this study support the classification of these three species as either biodiffusors or surficial modifiers. From a biogeochemical perspective it can therefore be anticipated that processes in the surface sediment layers are more affected by activities from this functional group of

30   fauna compared to processes that are mainly observed deeper in the sediment. Examples of such oxygen-including reactions are aerobic respiration and ammonium oxidation (nitrification). Further, the mixing of surficial layers, as well as the transformation of organic material due to feeding and excretion, will affect the composition and thereby fate of newly deposited material during mineralization (e.g. Aller, 1994).

[revised manuscript text omitted]

**1. Fauchald and Jumars (1979), 2. Queirós et al. (2013), 3. Sanders (1960), 4. Michaud et al. (2010), 5. Gilbert et al. (2007), 6. Duchêne and Rosenberg (2001), 7. Grémare et al. (2004) 8. Dufour and Felbeck (2003). *No information on the functional behavior of *Lipobranchius jeffreysii* and *Nuculana pernula* was found in the literature. These species were assumed to be functionally similar to *Scalibregma inflatum* and *Nuculana minuta*, respectively, due to their close affinity.**

**Table 2: Overview of experimental activities.**

| Day 2008 | Day 2009 | Activity |
|---|---|---|
| -54 | -45 | Field sampling of sediment |
| -43 to -46 | -34 to -36 | Aquaria were filled with sediment |
| -43 | -31 | $N_2$ treatment of the overlying water was initiated to remove possible fauna remaining in the sediment by asphyxiation |
| -36 | -27 | Air treatment of the overlying water was initiated to reoxygenate the surface sediment |
| -19 | -17 | Field sampling of benthic macrofauna |
| -12 to -13 | -8 to -9 | Addition of fauna to experimental aquaria |
| 0 | 0 | Addition of luminophores and start of tracer incubations |
| 0 to 14 | 0 to 14 | Sequential capturing of images |
| 14 to 16 | 14 to 15 | Termination of incubations |

**Table 3: Biomass and biovolume of benthic macrofauna. Each aquarium contained one individual, corresponding to an abundance of 460 ind. m$^{-2}$ (n=4).**

| Species | Biomass g aquarium$^{-1}$ (g m$^{-2}$) | Biovolume mL aquarium$^{-1}$ (mL m$^{-2}$) |
|---|---|---|
| *Glycera alba* | 0.16-0.23 (74-150) | 0.20-0.45 (92-210) |
| *Lipobranchius jeffreysii* | 0.21-0.46 (98-210) | 0.30-0.50 (140-230) |
| *Nephtys incisa* | 0.06-0.14 (28-65) | 0.10-0.15 (46-69) |
| *Scalibregma inflatum* | 0.13-0.20 (62-94) | 0.15-0.30 (69-140) |
| *Brissopsis lyrifera* | 0.19-0.24 (88-110) | 0.20-0.30 (92-140) |
| *Abra nitida* | 0.14-0.23 (65-110) | 0.20 (92) |
| *Nuculana pernula* | 0.59-0.64 (270-300) | 0.40 (180) |
| *Thyasira sarsii* | 0.056-0.078 (26-36) | 0.05-0.10 (23-46) |

**Table 4: Experimental variables that describe particle reworking. Relative amount (%) of luminophores added to the sediment surface that was observed below 0.5, 2 and 4 cm sediment depth (0.5cm, 2cm and 4cm), and the sum of distances between the shallowest and deepest penetration of luminophores observed during image analysis (Rug). Values are given as mean ± standard deviation.**

| Treatment | 0.5cm (%) | 2cm (%) | 4cm (%) | Rug (cm) |
|---|---|---|---|---|
| *Glycera alba* | 4.3 ± 3.5 | 0.4 ± 0.6 | 0.3 ± 0.6 | 930 ± 320 |
| *Lipobranchius jeffreysii* | 9.1 ± 9.3 | 2.9 ± 3.2 | 1.1 ± 1.3 | 1180 ± 980 |
| *Nephtys incisa* | 1.9 ± 2.5 | 0.4 ± 0.4 | 0.2 ± 0.2 | 600 ± 210 |
| *Scalibregma inflatum* | 1.2 ± 2.1 | 0.1 ± 0.2 | 0.0 ± 0.0 | 800 ± 250 |
| *Brissopsis lyrifera* | 4.3 ± 7.9 | 0.1 ± 0.1 | 0.0 ± 0.0 | 650 ± 130 |
| *Abra nitida* | 6.8 ± 9.6 | 0.0 ± 0.0 | 0.0 ± 0.0 | 580 ± 290 |
| *Nuculana pernula* | 24.0 ± 12.2 | 0.0 ± 0.0 | 0.0 ± 0.0 | 910 ± 450 |
| *Thyasira sarsii* | 1.6 ± 3.0 | 0.1 ± 0.0 | 0.1 ± 0.1 | 510 ± 70 |
| Control 2008 | 0.0 ± 0.0 | 0.0 ± 0.0 | 0.0 ± 0.0 | 310 ± 50 |
| Control 2009 | 0.1 ± 0.2 | 0.0 ± 0.0 | 0.0 ± 0.0 | 650 ± 80 |

[Figure]

5    **Fig. 1**

[Figure]

**Fig. 2**

[Figure]

**Fig. 3**

[Figure]

**Fig. 4**

[Figure]

**Fig. 5**

[Figure]

**Fig. 6**

[Figure]

**Fig. 7**

Revised manuscript

Supplement of the manuscript "*Functional classification of bioturbating macrofauna in marine sediments using time-resolved imaging of particle displacement and multivariate analysis*" (#bg-2016-411) by S. Lindqvist et al. for publication in Biogeosciences.

5    **Figure SM2: Relative distribution of fluorescent particles (luminophores) from side-view imaging of experimental aquaria following two-week incubations of sediment with eight faunal and two control treatments (filled black squares and solid line). Experimental distributions were evaluated by the gallery-diffusion model (filled gray boxes and dashed line). Displayed are the aquaria with the highest model error from each treatment. Units are in cm$^2$ y$^{-1}$ (Db) and y$^{-1}$ (r). Note the axis break in all treatments but that from *Nuculana pernula*.**

[Figure]

[Figure]

[Figure]

[Figure]

---

## Referee Comment (RC2) · Anonymous Referee #2 · 27 Dec 2016

Lindqvist et al. present a study that attempts to use high frequency measurements of particle dislocation in thin aquaria across a number of species to define functional groups. Often these functional group approaches use a set of a priori assumptions to define groups, then those groupings have often disappointed many when the actual measurements of some ecosystem function or sedimentary process is correlated to those groups. There are some areas of the paper that need revision prior to publication, and those general suggestions are listed below, which in my opinion, would vastly strengthen the manuscript.

[Figure]

**BGD**

One of the general strengths of the manuscript is the approach of measuring the effects of infauna on processes first, then using that data to define groups. The authors however miss some of the justification and prior work in this area, that I believe would help support their overall approach, the approach can be highlighted more clearly here by citing Gerino et al. (2003) and another similar perspective from Waldbusser and Marinelli (2009) with regards to functional groupings based on measured effects on processes versus observations of behavior. Along these same lines, I believe the paper could be more impactful if the authors relied on a broader range of literature that has tried to tackle the functional group issue in soft sediment systems (see papers by Pearson, Jumars, Hutchings, and Pearson dating back to the 1970's and up to the 2000's). Let's not forget the work previously done on this general area, and it would be useful for the authors to perhaps couch their functional grouping in the context of prior work as well. Ultimately will be ever have a common functional grouping scheme? Or is it all context dependent? I would also suggest the overall literature cited could be broadened a bit.

While I appreciate the statistical approach used, as it seems like an important way to let the data do the talking, I have some apprehension about how the PCA was applied, then interpreted, and possibly how well it can be extrapolated to other studies. I don't feel strongly enough to say it is incorrect, but a bit more information on correlated variables within the entire analysis would be helpful. It seems a bit strange to have variables that seem like they would be conveying the same thing, such as the different depths and a maximum penetration depth. It would strengthen the paper if the authors could provide a little more justification for the variable selection criteria, then also, how the variables that seem to come out of the PCA may fit into a broader understanding of the different impacts of infauna on particle displacement.

The authors recognize that most of the activity occurs within 48 hours of placing the luminophores on the sediment surface. What I cannot determine is whether that activity is included in the broader analysis, or if it is excluded as it represents a bulk sedi-

ment deposition event, which can vastly change behavior (see work by D'Andrea and Wheatcroft and also by Lohrer et al.). So, I would recommend that the authors more fully address this issue, as it seems it could be an important effector of their data, and thus possibly the outcomes of the analysis.

Finally, the figures could be perhaps conveyed in a bit more effective way? I also am unclear about the image of the system presented here. Is that a photograph that has had some filter applied to it? Why not just present the actual image? Some journals will not allow images that have been altered. If it is a line drawing, wow, the author could also be a graphic artist!

Overall, this is a nice study that should be ultimately published, the authors however need to do some more work to round out the paper a bit more, address some loose ends (as noted), and delve a bit deeper into the already published work.

---

## Author Comment (AC2) · 17 Jan 2017

Response from the authors to the interactive comment (RC2) on the manuscript "Functional classification of bioturbating macrofauna in marine sediments using time-resolved imaging of particle displacement and multivariate analysis" (#bg-2016-411) by S. Lindqvist et al. for publication in Biogeosciences.

We thank the reviewer for the comments provided that helped us further improve the revised version of the manuscript. We have to the best of our ability tried to meet the

comments by for example further clarifying and presenting (i) the concept of functional groups and the experimental approach in the introduction, (ii) the statistical method applied in materials and methods, and (iii) implications of results from the statistical method in the discussion section. We now hope that our manuscript will be accepted for publication as a regular article in Biogeosciences.

Yours sincerely

Stina Lindqvist /on behalf of all authors

Reviewer#2: ... One of the general strengths of the manuscript is the approach of measuring the effects of infauna on processes first, then using that data to define groups. The authors however miss some of the justification and prior work in this area, that I believe would help support their overall approach, the approach can be highlighted more clearly here by citing Gerino et al. (2003) and another similar perspective from Waldbusser and Marinelli (2009) with regards to functional groupings based on measured effects on processes versus observations of behavior. Along these same lines, I believe the paper could be more impactful if the authors relied on a broader range of literature that has tried to tackle the functional group issue in soft sediment systems (see papers by Pearson, Jumars, Hutchings, and Pearson dating back to the 1970's and up to the 2000's). Let's not forget the work previously done on this general area, and it would be useful for the authors to perhaps couch their functional grouping in the context of prior work as well. Ultimately will be ever have a common functional grouping scheme? Or is it all context dependent? I would also suggest the overall literature cited could be broadened a bit.

Response: We are pleased that our approach to couple definitions of functional groups with measured data is appreciated by the reviewer. As a response to the detailed comments by reviewer#1 we have clarified that the overall aim of the study was to "...quantitatively and qualitatively evaluate functionality of benthic macrofauna according to a wide spectrum of experimentally derived variables for particle transport". In

order to meet the comment by the reviewer related to the issue of further broaden the general concept of functional groups and highlight additional studies in this field, parts of the introduction were revised and additional references (Pearson (2001) and Wald-busser and Marinelli (2009)) added (p.2 l.6-9 and p.2 l.25-28 in the revised version#2). The references Gerino et al (2003) and Fauchald and Jumars (1979) are already cited (p.2 l.15 and e.g. p.11 l.29 in the original version). However, to emphasize previous efforts and approaches to quantify functional behavior of macrofauna, additional implications from e.g. these studies were added to the sections mentioned above.

Reviewer#2: While I appreciate the statistical approach used, as it seems like an important way to let the data do the talking, I have some apprehension about how the PCA was applied, then interpreted, and possibly how well it can be extrapolated to other studies. I don't feel strongly enough to say it is incorrect, but a bit more information on correlated variables within the entire analysis would be helpful. It seems a bit strange to have variables that seem like they would be conveying the same thing, such as the different depths and a maximum penetration depth. It would strengthen the paper if the authors could provide a little more justification for the variable selection criteria, then also, how the variables that seem to come out of the PCA may fit into a broader understanding of the different impacts of infauna on particle displacement.

Response: Text to further clarify the variables used during the PCA analysis has already been added to the revised version of the manuscript and in the response to comments by reviewer #1. To provide basic information on the PCA and correlations within variables as well as to clarify the interpretation of fig.6, text on the statistical presentation was added to the Materials and methods section (2.5 p.6 l.29-34). Additional information on variables was also added to the results section, p.9 l.24-25.

A paragraph on how the PCA analysis may fit into a broader understanding of particle transport by macrofauna was also added to p.10 l.17-19 in the revised version of the manuscript.

Co-linear variables in the dataset are normally not a problem for a PCA analysis. Quite in contrast, one of the reasons to apply a PCA is to reduce the number of variables and to transform co-linear or co-varying variables to a set of orthogonal principal components. Additional variables with slightly different selectivity often improve the discriminating capacity and robustness of the model. It is quite common to employ this framework to evaluate new or existing sets of variables, or to select the best variables for a specific purpose.

Reviewer#2: The authors recognize that most of the activity occurs within 48 hours of placing the luminophores on the sediment surface. What I cannot determine is whether that activity is included in the broader analysis, or if it is excluded as it represents a bulk sediment deposition event, which can vastly change behavior (see work by D'Andrea and Wheatcroft and also by Lohrer et al.). So, I would recommend that the authors more fully address this issue, as it seems it could be an important effector of their data, and thus possibly the outcomes of the analysis.

Response: Adding luminophores represents a bulk deposition of inorganic particles coated with fluorescent material. As presented by e.g. the 2D redistribution variable, effects on the tracer distribution from reworking were in general most pronounced within the first couple of days. This does not necessarily mean, however, that intensity of faunal reworking was highest during this time period. As observed in e.g. the variable for daily transport, which quantifies the amount ($\approx$ number) of particles relocated over 24 hours (fig. 5), patterns of particle transport by fauna seemed rather constant with time. The temporal effects of luminophore displacements are not directly evaluated as a factor by the PCA model, although individual variables vary with time. The introduction of experimental variables in multivariate analysis to capture temporal effects during investigations of functionality of fauna was considered outside the scope of the present study.

Reviewer#2: Finally, the figures could be perhaps conveyed in a bit more effective way? I also am unclear about the image of the system presented here. Is that a photograph

that has had some filter applied to it? Why not just present the actual image? Some journals will not allow images that have been altered. If it is a line drawing, wow, the author could also be a graphic artist!

Response: It is from the comments provided difficult to respond and understand exactly how the reviewer thinks we should revise the figures. No changes made.

The image of the experimental system (Fig. 1) is a photograph. It was modified in Adobe Photoshop and Gimp to make the setup clearer. For example, contrast was increased and blurry areas clarified manually. As was also pointed out in the figure captions, it is a schematic overview and not the original image. No changes made in the manuscript.

Please also note the supplement to this comment:
http://www.biogeosciences-discuss.net/bg-2016-411/bg-2016-411-AC2-supplement.pdf

**Supplement:**

[revised manuscript text omitted]

25   2006). Further, a concept of functional groups based on how sediment properties and characteristics are affected by faunal reworking activities, rather than by patterns of faunal behavior per se, may provide an experimental tool to relate activities by benthic fauna to an observed biogeochemical response (Gerino et al., 2003, Waldbusser and Marinelli, 2009). Species-specific reworking traits have been demonstrated to be important for ecosystem function *in situ*, in laboratory, and *in silico* experiments (Lohrer et al., 2004;Solan et al., 2004a;Norling et al., 2007) and the

30   impact of rates and magnitude of particle displacements by functional traits of benthic macrofauna has been the focus of several studies (e.g. Gilbert et al., 2007;Braeckman et al., 2010;Hedman et al., 2011).

Four major reworking modes based on shared patterns of particle reworking have so far been defined; up- and downward conveyors, regenerators and biodiffusors (François et al., 1997). The upward conveyors are described as

35   vertically oriented deposit feeders that ingest sediment at depth, and defecate at the sediment surface (Rhoads, 1974;Robbins et al., 1979). Such feeding mode causes an advective transport of sediment from depth to the sediment surface. Similarly, the downward conveyors are vertically oriented species that feed at the surface and defecate deep

in the sediment (Smith et al., 1986). The regenerators excavate burrows by transferring sediment to the surface. The burrows are subsequently filled by either surface sediment infilling or collapsing of the burrow walls (Benninger et al., 1979;Gardner et al., 1987). Biodiffusors are organisms with activities that result in a constant diffusive-like transport of sediment, i.e. particles are transported in a random manner over short distances (Boudreau,

5    1986a;Robbins et al., 1979;François et al., 1997). In addition to these four main groups, the gallery diffusors are surface-active species that create galleries of burrows. Rapid, advective transport of particles occurs in the burrow system while the upper sediment layer is reworked in a diffusive manner. Initially suggested as a separate group (François et al., 2002), it has also been described as a subgroup of the biodiffusors, together with the epifaunal biodiffusors and surficial biodiffusors (Kristensen et al., 2012). In the classification scheme proposed by Solan et al.

10   (2004a) epifauna form a separate group as do surficial modifiers, fauna with activities restricted to the sediment layer immediately below the sediment surface (<1-2 cm sediment depth).

Classification of benthic macrofauna from patterns of particle reworking is mainly based on laboratory experiments including one or a few species normalized to either constant biomass or abundance (e.g. Gilbert et al., 2007;François et al., 2002). Mode of reworking is then generally determined by a few reworking variables modeled from the

15   vertical redistribution of a particle tracer (Wheatcroft et al., 1990;Delmotte et al., 2008). Recently, functional classification was provided for more than 1000 benthic invertebrates in order to enable calculations of the community bioturbation potential (BPc) (Queirós et al., 2013). Due to a paucity of data on the reworking behavior for many bioturbating species, classifications were not only based on published material but also expert knowledge and genetic similarity between species.

20   Quantitative comparisons and classifications of particle reworking activities by benthic macrofauna provide a versatile tool to understand and predict feedbacks within and between macrofaunal communities, as well as between advective transport of particles by fauna and sediment biogeochemistry in natural environments.

The aim of the study was to quantitatively and qualitatively evaluate functionality of benthic macrofauna according

25   to a wide spectrum of experimentally derived variables for particle transport. Patterns of particle relocation were experimentally quantified and modeled by multivariate analysis. 
[revised manuscript text omitted]
 particles compared to the initial location of luminophores. Only the last value in the time series for 2D redistribution was used in the statistical analysis. The daily transport was calculated by subtracting the binary image obtained at time t ($M_t$), with the binary image obtained the previous day ($M_{t-24\text{ h}}$) and summing the absolute values of the resulting matrix. 2D redistribution and daily transport express the number of pixels that represent particle transport. The number of pixels was converted to area units ($cm^2$) for purpose of comparison. The transport rate of particles, slightly modified from that described in Lindqvist et al. (2013), was calculated from the sum of the daily transport ($\Sigma T_{daily}$) divided by the number of experimental days ($\Sigma T_{daily}/14$).

The 2D variable rugosity (Rug; Murray et al., 2014) was extracted from the last image in the time series.  The distance between the shallowest and the deepest luminophore was calculated for each column in the image. Rugosity was calculated as the sum of these distances (Murray et al., 2014).

The pixels of vertical layers including luminophores were summed and assigned to the midpoint of each layer to reduce the 2D image to a 1D vertical profile. From these profiles, variables describing the relative fraction (%) of luminophores transported beneath 0.5 (0.5cm), 2 (2cm) or 4 cm (4cm) were extracted. 1D tracer distributions were also used to quantitatively evaluate the particle transport by the gallery-diffusion model, which approximates the diffusive mixing by the biodiffusion coefficient (Db) and the rapid transport over long distances by the non-local transport coefficient, r (François et al., 2002). The model is time-and space dependent and employs ordinary differential equations to minimize a weighted sum of squared difference between the observed and modeled tracer concentrations with sediment depth.

The maximum penetration depth of luminophores (MPD) was calculated from the deepest pixel row containing luminophores (Maire et al., 2006). To exclude noise, the limit of quantification was set to 3 pixels in 2008 and 8 pixels in 2009. The limit was determined from the camera response in deep sediment layers of the control cores with no luminophores (i.e. background). The sediment depth at which animal activity was observed from photos of the aquaria walls taken under normal light conditions was also determined (Burrow).

**2.5 Statistical analysis**

Principal component analyses (PCA) were performed in Simca (v. 14.0, Umetrics AB, Sweden) to identify groups with similar particle reworking behavior and to evaluate the most relevant reworking variables to be included in models of functional classification. PCA is a multivariate projection method designed to reduce dimensionality and to identify patterns in a data set. Variables are evaluated and patterns in the data set are revealed by ordination of objects in lucid 2D plots. Positively correlated variables contribute similar information and are grouped together. Negatively ('inversely') correlated variables are positioned on opposite sides of the plot origin, in diagonally opposed quadrants. The further away from the plot origin a variable is, the stronger impact that variable has on the model. Variables are converted to principal components that form an orthogonal basis for the space of the data. Each principal component is a linear combination of the original variables orthogonal (uncorrelated) to the preceding component. Prior to analysis, variables were assessed by graphical exploration of box plots (Quinn and Keough, 2002). Where appropriate, variables were log(x) or log(x+1) transformed in order to fit the requirements for PCA.

[revised manuscript text omitted]

5    **Fig. 1**

[Figure]

**Fig. 2**

[Figure]

**Fig. 3**

[Figure]

**Fig. 4**

[Figure]

**Fig. 5**

[Figure]

**Fig. 6**

[Figure]

**Fig. 7**